# Convective Gusts Nowcasting Based on Radar Reflectivity and a Deep Learning Algorithm

Haixia Xiao[1], Yaqiang Wang[2,*], Yu Zheng[1,*], Yuanyuan Zheng[1], Xiaoran Zhuang[1,3], Hongyan Wang[2], and Mei Gao[2]

[1]Key Laboratory of Transportation Meteorology of China Meteorological Administration, Nanjing Joint Institute for Atmospheric Sciences, Nanjing 210041, China
[2]Chinese Academy of Meteorological Sciences, Beijing 100081, China
[3]Jiangsu Meteorological Observatory, Nanjing 210041, China
[*]These authors contributed equally to this work.

**Correspondence:** Yangqiang Wang (yqwang@cma.gov.cn), Yu Zheng (zhengyu@cma.gov.cn)

**Abstract.**

Convective wind gusts (CGs) are usually related to thunderstorms, and they may cause great structural damage and serious hazards, such as train derailment, service interruption, and building collapse. Due to the small-scale and nonstationary nature of CGs, reliable CGs nowcasting with high spatial and temporal resolutions has remained unattainable. In this study, a novel nowcasting model based on deep learning - namely, CGsNet - is developed for 0-2 hour lead times of quantitative CGs nowcasting, achieving minute-kilometer-level forecasts. CGsNet is a physics-constrained model established by training on large corpora of average surface wind speed (ASWS) and radar observations, it can produce realistic and spatiotemporally consistent ASWS predictions in CGs events. By combining the gust factor (1.77, the ratio of the observed peak wind gust speed (PWGS) to the ASWS) with the ASWS predictions, the PWGS forecasts are estimated with a spatial resolution of $0.01° \times 0.01°$ and a 6-minute temporal resolution. CGsNet is shown to be effective, and it has an essential advantage in learning the spatiotemporal features of CGs. In addition, quantitative evaluation experiments indicate that CGsNet exhibits higher generalization performance for CGs than the traditional nowcasting method based on numerical weather prediction model. CGs nowcasting technology can be applied to provide real-time quantitative CGs forecasts.

## 1 Introduction

Convective wind gusts (CGs) are nontornadic, straight-line winds (Mohr et al., 2017; Yu and Zheng, 2020). They predominantly occur in Eastern China in the warm summer months (Yang et al., 2017), and they are usually associated with thunderstorm clouds, in particular squall lines and supercells, which generate conditions conducive to the appearance of whirlwinds and squalls (Kolendowicz et al., 2016). Severe convective gusts are caused either by mesoscale cold pools associated with horizontal pressure gradients large enough to produce high wind speeds in the absence of strong downdrafts or by local-scale downbursts that create strong divergent horizontal winds near the ground (Wakimoto, 2001). These severe convective gusts may cause

considerable damage to communication, transport, buildings and other structures, and even the health and life of humans in many parts of the world (Mohr et al., 2017; Wang et al., 2020).

CGs as defined by the China Meteorological Administration (CMA) are usually recognized as a PWGS $\geq$ 17.2 m/s caused by severe atmospheric convection (Yu and Zheng, 2020). Compared with large-scale wind storms, many observations show
that convectively induced storm events can reach higher wind speeds (Mohr et al., 2017). For example, the CGs observed in 1983 during a microburst in the United States had a peak gust of 67 m/s (Fujita, 1990), for which wind speeds are comparable to those of an F3 tornado based on damage assessments. Thus, forecasting the occurrence of CGs in advance is valuable for reducing the risk and threat of damaging wind events.

Currently, accurate CGs nowcasting is still a challenging issue in operational meteorology despite its severe impact (Ray,
2015; Stensrud et al., 2009). This is primarily for several reasons. One is the small-scale and nonstationary nature of CGs. Due to the rapid evolution of fine-scale convective systems and their complicated interactions with environmental features, it is difficult for mesoscale numerical models or statistical models to capture and forecast CGs. In addition, the sparsity of wind observations is a factor that hinders the development and verification of gust nowcasting models. The most frequently cited record-breaking wind speeds are usually associated with wind gust values. This refers to a maximum momentary (2-3 sec.)
wind speed that exceeds the 2-minute mean wind speed by at least 5 m/s (WMO, 2010). At present, wind gusts between full hours of observation have been recorded in Eastern China, which makes it possible to determine the actual maximum wind gusts/peak wind gusts (PWGS) for each hour. Obviously, these existing meteorological observation networks cannot record fine-scale CGs accurately (Kahl et al., 2021; Mohr et al., 2017), and a significant portion of the highest wind gusts may be missing.

Many previous studies have mainly focused on potential severe convective weather (SCW) forecasting (McNulty, 1995; Doswell et al., 1996) or the possibility of classified SCW forecasting (Zhou et al., 2019; Lagerquist et al., 2017), while quantitative CGs nowcasting has rarely been reported. Currently, some studies focus on wind gust forecasting by using physically based methods, empirical results and tuning, statistical analysis or machine learning methods (Kahl, 2020; Kahl et al., 2021; Lei et al., 2009; Sheridan, 2018; Nerini et al., 2014; Chaudhuri and Middey, 2011). Although some progress has been made in
these studies, their performance on wind gust nowcasting (0-2 h) is poor (Wang et al., 2016), especially in terms of spatial and temporal resolution. For example, the wind forecasted by Integrated Nowcasting through Comprehensive Analysis (INCA) (Haiden et al., 2011) or the newly developed meteorologically stratified gust factor (MSGF) model (Kahl et al., 2021) has a 1-hour temporal resolution, which is not sufficient for describing the evolution of wind gusts. Most importantly, few gust forecasting models have been evaluated for effectiveness in CGs nowcasting.

Recent advances in deep learning (Yang et al., 2017), especially recurrent neural networks (RNNs) and convolutional neural networks (CNNs), have provided some useful insights into the meteorological field (Duan et al., 2021; Sadeghi et al., 2020). Many approaches based on RNNs and CNNs have been applied successfully to short-term weather forecasting due to their ability to capture temporal and spatial variations in image features, e.g., short-term drought (Danandeh Mehr et al., 2022) and quantitative precipitation forecasts (Shi et al., 2015). An evaluation of the applications indicates that the deep learning solutions
can model complex nonlinear systems and better extract advanced features.

Complex physical processes and dynamic characteristics are often involved in convective systems at small spatial and temporal scales (Doswell, 2001). It is critical to fully extract convective characteristics automatically to improve the forecast accuracy, and deep learning provides a practical tool for this purpose. In particular, Zhou et al. (2019) developed a forecasting solution for SCW using a CNN, producing probabilistic forecasting products. Their results showed that deep learning methods achieve better performance than traditional forecasting methods. Thus, as it is a similar problem, deep learning may have considerable application potential in CGs nowcasting, and it is promising to test CGs forecasting using deep learning methods, which has rarely been documented before.

The objective of this study is to achieve quantitative nowcasting of CGs speed in eastern China using the deep learning method, in principle down to the minute-level timescale. Doppler weather radar is an important device for observing and forecasting CGs because of abundant weather information in radar reflectivity (e.g., bow echo and hook echo) (Holleman, 2001; Yuan et al., 2018; Yu and Zheng, 2020). Thus, in this study, we aim to forecast quantitative CGs based on observed wind and radar data, which has rarely been considered in previous studies. We also compare the nowcasting performance of the deep learning method with that of the INCA approach. Notably, we concentrate on the nowcasting and evaluation of the average surface wind speed (ASWS) and PWGS, which are both key factors that reflect the strength and influence of CGs. We believe this work will have significant application potential in small-scale hazard assessment and alerting of threats to the economy and human activities.

## 2 Data

The study area is in Eastern China (Figure 1), including Jiangsu Province and its adjacent area. The latitudinal and longitudinal ranges of the study area are 30.4°N–35.2°N and 116.33°E–121.93°E, respectively. Considering that CGs associated with thunderstorms occur predominantly in the warm season in eastern China (Yang et al., 2017), April to September of each year was selected as the period of the experiments in this study.

### 2.1 Wind data

For the ground-truth reference, the ASWS data comprise in situ observations with a 5-minute time interval, which is the 2-minute average of the 3-second wind measured using automatic weather stations (AWSs) (Figure 1b) at a height of 10 meters above the ground. Note that there are limited wind observation data available on the seacoast/over the sea, and as a result, this study focuses on gust forecasting in the land region of eastern China. To match the spatiotemporal resolution of the observed radar reflectivity mosaics (RMOSs), the ASWS data were first interpolated to 6 minutes using linear interpolation. Inverse distance-weighted (IDW) interpolation has the advantages of being fast to calculate and easy to understand, and it is suitable for eastern China, where the automatic stations are dense and evenly distributed. Thus, the IDW interpolation method was employed to regrid the ASWS data to a spatial resolution of 0.01° × 0.01° by using the four nearest stations within 15 km with a power of 2 (the details of the parameter choices can be found in the supplement). The wind observation data on the seacoast and over the sea was set to 0. Subsequently, during the training process, we masked the wind data on the seacoast

and over the sea by setting the weight of the loss function to 0. This ensures that the sea areas do not contribute to the backpropagation optimization. However, the values in the sea areas still participate in the convolutional operations during the forward propagation process, which could lead to a slight underestimation of the ASWS or PWGS values in areas close to the seacoast. In this study, the interpolated wind data are assumed to be close to the true observed wind field.

ASWS data from 2016 to 2022 were considered. To further select ASWS data associated with CGs, the samples selection need meet two concurrent principles: 1) more than 2% of stations recorded ASWS > 10.8 m/s; and 2) the precipitation at more than 5% of the stations within an hour was greater than 0.1 mm.

In addition, observed peak wind gust speed (PWGS) from 2021 to 2022 was used in this study for subsequent gust factor calculation and model evaluation; it is the maximum 3-second average gust recorded over 1 hour by the AWS. Samples from PWGS were taken from typical CGs events recorded by the meteorological bureau.

## 2.2 Radar reflectivity

Weather radar provides a highly detailed representation of the spatial structure and temporal evolution of thunderstorms over a large area (Knupp, 1987). Weather radar data may also indicate the time and location of the CGs that occurred (Holleman, 2001). In particular, the constant-altitude radar RMOSs at 3 km are less susceptible to ground clutter compared to radar data at other altitudes. This allows them to provide continuous and comprehensive coverage of the study area, as well as a more accurate representation of storm structure. Thus, these constant-altitude radar RMOSs at 3 km were used as auxiliary data along with observed wind data to forecast the speed of CGs. The RMOS data is from 10 weather radar stations in eastern China (Figure 1a). The type of radar sensor is the CINRAD/SA Doppler weather radar (Sheng et al., 2006). The RMOS is acquired

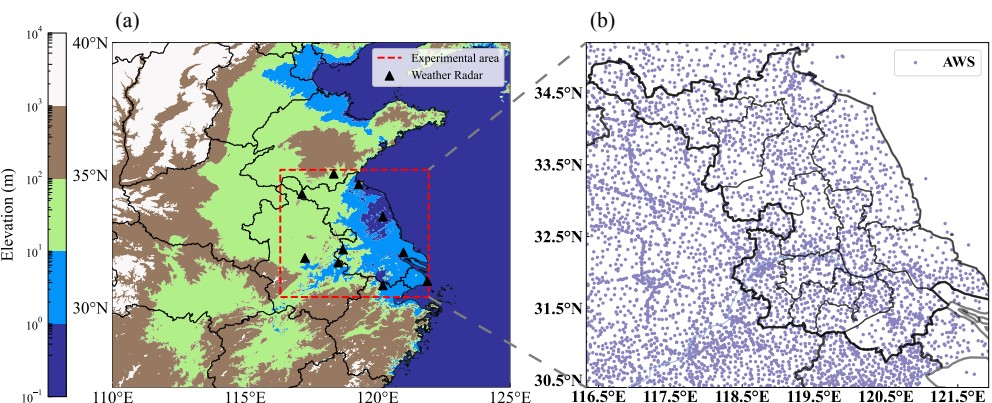

**Figure 1.** The selected study area: the red dotted line in (a) represents Eastern China (30.4°N–35.2°N, 116.33°E–121.93°E); the black triangles in (a) represent the locations of the weather radar; and the purple dots in (b) are the locations of the AWSs that currently measure ASWS and PWGS.

through the operational Doppler weather radar 3-D digital mosaic system (Wang et al., 2009), which was developed by the

Chinese Academy of Meteorological Sciences. This system determines the RMOS by projecting the base reflectivity factor onto equal longitude and latitude coordinates. The time interval of the RMOS data is 6 minutes, the spatial resolution is 0.01°, and the grid data are arranged from west to east and north to south.

## 2.3 Workflow

The workflow of this study is illustrated in Figure 2. First, the RMOS data were preprocessed, and the ASWS data were

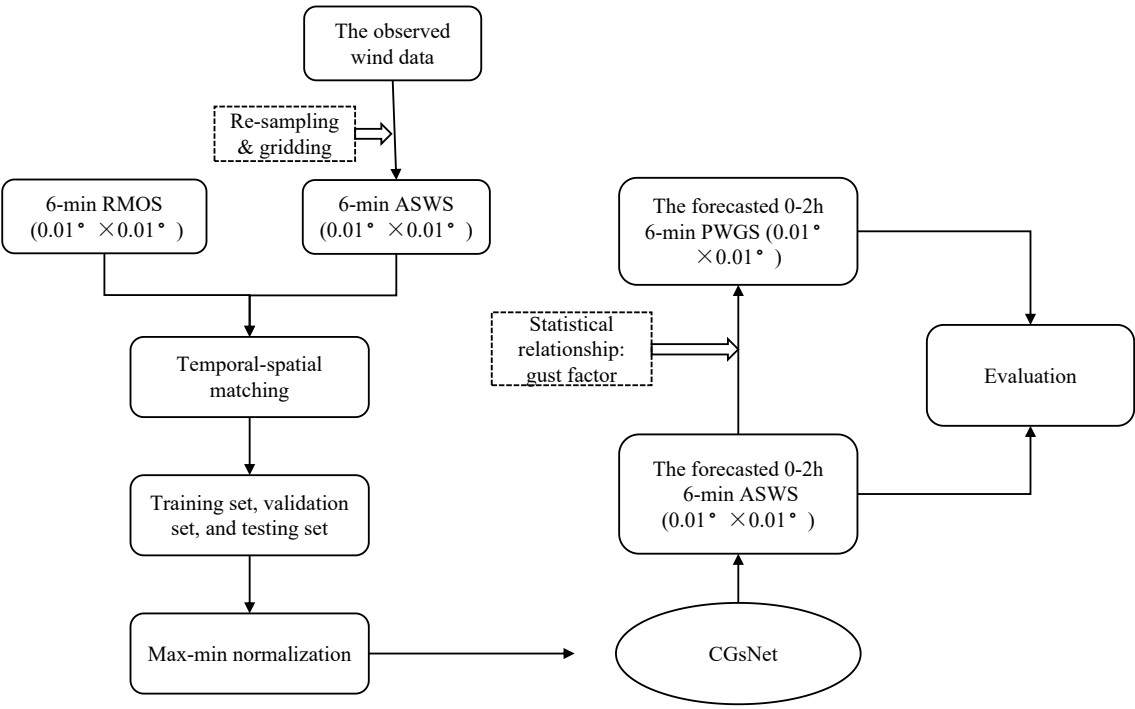

**Figure 2.** Workflow of deep learning model-based CGs nowcasting.

interpolated and regridded as mentioned above. Second, the spatial-temporal resolutions of the RMOS and ASWS samples were matched, and the time of the RMOS samples was determined by the selected ASWS samples. Third, the dataset was divided into three parts (training, validation and testing datasets) according to the observation time. Fourth, the ASWS data were clipped to between 0 and 35 m/s and subsequently normalized to the range [0.0, 1.0] through min-max normalization. Similarly, the RMOS values were clipped to between 0 and 70 dBZ and subsequently normalized to between 0 and 1. Fifth, the ASWS and RMOS data in the past 60 minutes were used as input data to train and validate the deep learning nowcasting model, and the outputs were the forecasted ASWS with lead times of 6-120 minutes ahead, with a spatial resolution of 0.01° × 0.01° and a 6-minute temporal resolution. Sixth, the performance of the deep learning model in forecasting ASWS was evaluated

by different statistical variables. Seventh, the mean gust factor (GF) was determined by comparing the observed PWGS and ASWS. Subsequently, the PWGS forecasts were estimated by multiplying the GF by the forecasted ASWS. Finally, the PWGS forecasts were compared with INCA results, and two cases were also showed to demonstrate the accuracy of PWGS forecasts.

## 3 Model and settings

### 3.1 Model architecture

We started by adapting the network structure proposed by Guen and Thome (2020b), namely, PhyDNet, which is a sequence-to-sequence network. Unlike traditional networks, PhyDNet steers the learning process toward identifying physically consistent solutions by introducing an appropriate inductive bias (Karniadakis et al., 2021), that is, implicitly embedding prior knowledge in the network architecture and satisfying a given set of physical laws. It has a two-branch deep architecture, i.e., the newly proposed recurrent physical cell (PhyCell) and ConvLSTM. Specifically, PhyCell is inspired by data assimilation techniques for 130 performing partial differential equation (PDE)-constrained prediction in latent space. It explicitly disentangles PDE dynamics from unknown complementary information and allows for a generic class of linear PDEs through varying differential orders, e.g., wave equations. In PhyDNet, the physical knowledge is represented by PDEs, which can enforce physical constraints for future image prediction well. PhyDNet has been successfully employed in many fields, e.g., solar irradiance forecasting using fisheye images (Guen and Thome, 2020a) and nitrogen prediction (Jahanbakht et al., 2022).

The attention mechanism can focus on the important information among a large amount of input information and ignore most of the unimportant information (Niu et al., 2021). Thus, an attention mechanism is applied to improve the performance of PhyDNet for nowcasting CGs in this study. Here, the PhyDNet with an attention mechanism for CGs nowcasting is called CGsNet, and its architecture is illustrated in Figure 3. An input sequence $\{x_{t-K}^{a,r}, \ldots, x_t^{a,r}\} \in \mathbb{R}^{K \times n \times l \times c}$ including ASWS ($a$) and RMOS ($r$) with $K$ time steps, spatial size $n \times l$, and $c$ channels is projected into $\{h_{t-K}^{E}, \ldots, h_t^{E}\}$ by the convolutional units 140 and processed by the recurrent block unfolded in time. This forms a sequence-to-sequence architecture suited for multistep prediction, outputting $T$ future ASWS ($a$) and RMOS ($r$) predictions $\{\hat{x}_{t+1}^{a,r}, \ldots, \hat{x}_{t+T}^{a,r}\}$. For time t, we learn a weight vector $\langle \alpha_{tk}, \ldots, \alpha_{t1} \rangle$ for $K$ hidden states $\{h_{t-K}^{m}, \ldots, h_{t-1}^{m}\}$, where the weight $\alpha_{tk}$ can be interpreted as the relative importance of the $k$-th $h^m$. Each weight $\alpha_{tk}$ is computed by taking $h_{t-k}^{E}$ as input, followed by a softmax operation. Then, we perform summation over the weighted hidden states to obtain an attention representation $A_t$, which will be incorporated into the decoder for 145 prediction. The key equations for generating the attention representation $A_t$ for the $t+1$-th output prediction are summarized

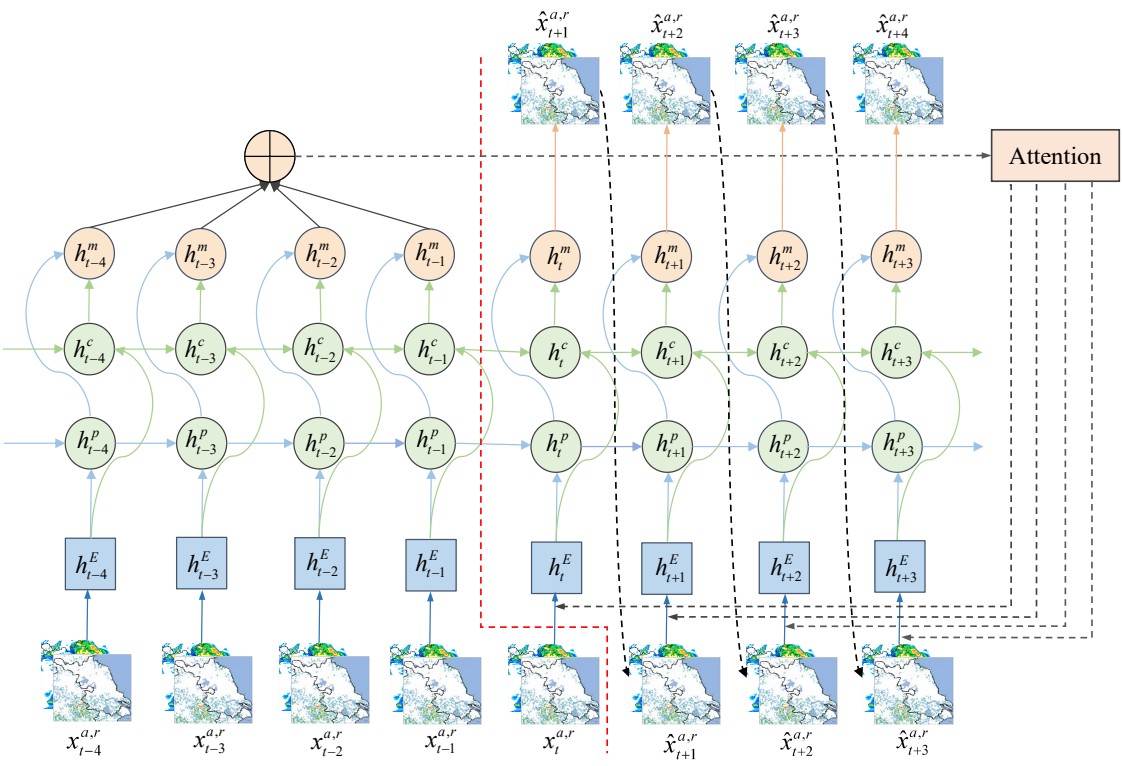

**Figure 3.** Illustration of CGsNet. The encoder is to the left of the dotted red line, and the decoder is to the right. $x_i^{a,r}$ and $\hat{x}_i^{a,r}$ are the observed and forecasted ASWS/RMOS fields, respectively. $h_i^E$ indicate the input tensors calculated by the convolution units. $h_i^c$ and $h_i^p$ indicate the hidden states of ConvLSTM and PhyCell, respectively. $h_i^m$ represents the hidden state that combines the values from $h_i^c$ and $h_i^p$.

as follows.

$$s_{tk} = W * h_{t-k}^E + b, \quad \forall k \in [1, K] \tag{1}$$

$$\alpha_{tk} = \frac{\exp(s_{tk})}{\sum_{k=1}^{K} \exp(s_{tk})}, \quad \forall k \in [1, K] \tag{2}$$

$$A_t = \sum_{k=1}^{K} \alpha_{tk} h_k^m \tag{3}$$

$$h_{t+1}^m = PhyCell([h_t^E, A_t]) + ConvLSTM([h_t^E, A_t]) \tag{4}$$

where '$*$' denotes the convolution operator and the kernel matrices $W$ and biases $b$ are parameters to be learned. The high-level representation $h_{t+1}^m$ is then fed into the deconvolutional units, and $\hat{x}_{t+1}^{a,r}$ is calculated. Similarly, by inputting $\hat{x}_{t+1}^{a,r}$ into the decoder, combined with the attention mechanism, the forecasted $\hat{x}_{t+2}^{a,r}$ can be calculated. In this cycle, we can obtain the ASWS

and RMOS forecasts in the future, i.e., $\{\hat{x}_{t+1}^{a,r}, \ldots, \hat{x}_{t+T}^{a,r}\}$. The effect of the attention mechanism operation is to find the most significant historical information from the input sequences, e.g., when predicting $\hat{x}_{t+1}^{a,r}$, the attention module can assign weights for $\{h_{t-K}^m, \ldots, h_t^m\}$. This can better model both short-term and long-term dependencies. The detailed convolutional encoder and decoder, the calculation process of PhyCell and ConvLSTM, and the feature maps of hidden states can be found in the Supplement. Note that we just focus on analyzing the forecasting results of ASWS, and the forecasted RMOS is a secondary result.

## 3.2 Experimental settings

Here, we aim to forecast 0-2 hour CGs; thus, 10 historical ASWS and RMOS grid fields $\{x_1^{a,r}, \ldots, x_{10}^{a,r}\} \in \mathbb{R}^{10 \times 480 \times 560 \times 2}$ were taken as the inputs, and the subsequent 20 ASWS and RMOS grid fields $\{x_{11}^{a,r}, \ldots, x_{30}^{a,r}\} \in \mathbb{R}^{20 \times 480 \times 560 \times 2}$ were the ground-truth outputs. The ASWS and RMOS grid fields from 2016 to 2020 were used for model's training, and the fields from June to September 2021 and April to May 2021 were employed for ASWS validation and testing, respectively. The fileds from May to July 2022 were used for PWGS testing. Correspondingly, we obtained 15,629 CGs samples, including a training set of 13,155 samples, a validation set of 1,236 samples, and the testing sets of 1,184 and 54 samples for ASWS and PWGS.

Regarding the training settings, Adam (Kingma and Ba, 2014) was used as the optimizer, and the batch size was set as 2. The learning rate was initially set to 0.001 and then multiplied by 0.3 if the loss did not decrease when the model was trained for 2 epochs. The model was trained for a total of 50 epochs, and the model weight with the minimum loss on the validation set was selected and saved. Then the evaluation experiments were implemented on the testing set. Detailed hyperparameter choices can be found in the supplement. All experiments in this study were conducted using Python and PyTorch and executed on 4 NVIDIA GeForce RTX 3090 GPUs with the CUDA-11.1 library.

## 3.3 Loss function

The loss function is used to adjust the weights of the deep learning model, and it is an instructive function that guides how to improve the weight coefficients. Most previous studies used the mean absolute error (MAE) loss or mean squared error (MSE) loss as the default to train their models (Tran and Song, 2019; Veillette et al., 2020; Franch et al., 2020). However, these losses will result in smoothing effects in the predicted fields, since they may make a good assumption about the global similarity of the two fields, meaning that the field's details may be lost (Tran and Song, 2019; Franch et al., 2020). Notably, with increasing forecast time, the problem of underestimating high values becomes more obvious, such as underestimating heavy rain or radar reflectivity. To address these problems and make the predictions sharper and more realistic, the weighted MAE loss (Shi et al., 2017) was adopted in this study. The weights for the MAE loss function are shown in Table 1 (the further details on parameter choices are available in the supplement). These weights are determined based on the six ASWS and RMOS value ranges. The weighted MAE loss is designed to ameliorate the problem of data imbalance caused by the scarcity of strong gust data, as it gives a large weight to strong gusts and a small weight to weak gusts. Thus, this setting has a better effect on forecasting CGs

than loss without weights. The weighted loss function can be calculated as follows.

$$loss = \frac{1}{N} \sum_{n=1}^{N} \sum_{i=1}^{480} \sum_{j=1}^{560} w_{n,i,j} \left| y_{n,i,j}^{a} - \hat{y}_{n,i,j}^{a} \right| + \frac{1}{N} \sum_{n=1}^{N} \sum_{i=1}^{480} \sum_{j=1}^{560} w_{n,i,j} \left| y_{n,i,j}^{r} - \hat{y}_{n,i,j}^{r} \right| \tag{5}$$

where N is the number of samples, and $i$ and $j$ are the width and length of the ASWS/RMOS, respectively. Moreover, $y_{n,i,j}^{a}$, $y_{n,i,j}^{r}$, $\hat{y}_{n,i,j}^{a}$, and $\hat{y}_{n,i,j}^{r}$ are the observed ASWS, observed RMOS, forecasted ASWS and forecasted RMOS values, respectively. $w_{n,i,j}^{a}$ and $w_{n,i,j}^{r}$ denote the weights of the ASWS and RMOS values, respectively.

**Table 1.** The weights of the ASWS and RMOS in the MAE loss.

| Wind (m/s) | Weights | RMOS (dBZ) | Weights |
|---|---|---|---|
| y ≤ 5.5 | 0.5 | y ≤ 15 | 0.5 |
| 5.5 ≤ y ≤ 8.0 | 1 | 15 ≤ y ≤ 25 | 1 |
| 8.0 ≤ y ≤ 13.9 | 2 | 25 ≤ y ≤ 35 | 2.5 |
| 13.9 ≤ y ≤ 17.2 | 10 | 35 ≤ y ≤ 45 | 5 |
| 17.2 ≤ y ≤ 20.8 | 20 | 45 ≤ y ≤ 50 | 10 |
| y ≥ 20.8 | 30 | y ≥ 50 | 15 |

## 3.4   Model evaluation

To quantify the capabilities of CGsNet, the critical success index (CSI), probability of detection (POD), false alarm rate (FAR), bias score, and Heidke skill score (HSS) were implemented for evaluating the CGsNet forecasts. The following equations describe these indices.

$$CSI = \frac{TP}{TP + FP + FN} \tag{6}$$

$$POD = \frac{TP}{TP + FN} \tag{7}$$

$$FAR = \frac{FP}{TP + FP} \tag{8}$$

$$BIAS = \frac{TP + FP}{TP + FN} \tag{9}$$

$$HSS = \frac{(TP \times TN - FN \times FP)}{(TP + FN) \times (FN + TN) + (TP + FP) \times (FP + TN)} \tag{10}$$

where TP, TN, FN, and FP are the numbers of true positives, true negatives, false negatives, and false positives, respectively.
Moreover, the MAE and RMSE are also used for model evaluation, and are calculated as follows.

$$MAE = \frac{1}{Q} \sum_{i=1}^{Q} |y_i - \hat{y}_i| \qquad (11)$$

$$RMSE = \sqrt{\frac{1}{Q} \sum_{i=1}^{Q} (y_i - \hat{y}_i)^2} \qquad (12)$$

where Q is the number of predicted ASWS pixel values. $y_i$ and $\hat{y}_i$ represent the observations and forecasts, respectively. Note that the value of BIAS is greater than 0, and the closer to 1 the value is, the better. The remaining indices range between 0 and 1. An excellent model should present high CSI, POD, and HSS values but low FAR, MAE, and RMSE values.

The bootstrapping method is used to generate the 95% confidence intervals for the mentioned indices. Confidence interval (CI) helps to provide a measure of the uncertainty associated with the results and allows for a better understanding of the range of possible values for the indices. Besides, the use of bootstrapping allows us to estimate the variability of the indices and helps to ensure that the results are robust and reliable.The detailed calculation process can be found in the supplement.

In addition, the baseline methods, PhyDNet and INCA (Haiden et al., 2011), are compared with CGsNet in forecasting ASWS and PWGS in CGs, respectively. INCA is a nowcasting system developed with complex terrain adaptability at the Austria National Weather Service. In this study, the nowcasts start with a three-dimensional analysis based on a first guess obtained from the output of the numerical weather prediction model (here, the PWAFS model (Li et al., 2016) is used), with observation corrections superimposed. It can provide wind forecasts, for which the update frequency (temporal resolution) is 1 h.

In particular, we concentrate on evaluating ASWS at thresholds of 8.0-13.9 m/s (e.g., ASWS > 8.0 m/s or ASWS > 13.9 m/s) and PWGS at thresholds of 10.8-20.8 m/s in CGs events, which has broad value for disaster warning in meteorological operations and aviation applications.

## 4 Results

### 4.1 Model performance on ASWS nowcasting

To validate and evaluate the effectiveness of the CGsNet on forecasting ASWS in CGs, the baseline model, PhyDNet, was used for the comparison. The evaluation results of CGsNet and PhyDNet on the CSI, HSS, POD, MAE, and RMSE are given in Table 2 for the investigated nowcasting lead time. The thresholds at 8.0 m/s, 10.8 m/s and 13.9 m/s of ASWS are evaluated, and the results reveal that CGsNet outperformed PhyDNet on these criteria. In detail, for an ASWS threshold of 8.0 m/s, CGsNet has better performance with HSS = 0.54 (95% CI: 0.47; 0.61), POD = 0.59 (95% CI: 0.51; 0.66), MAE = 1.6 (95% CI: 1.46; 1.80) m/s, and RMSE = 2.26 (95% CI: 2.00; 2.54) m/s than PhyDNet (HSS = 0.52 (95% CI: 0.44; 0.59), POD = 0.55 (95% CI: 0.46; 0.62), MAE = 1.71 (95% CI: 1.55; 1.91) m/s, and RMSE = 2.40 (95% CI: 2.14; 2.68) m/s). When the threshold is raised to 10.8 m/s, the forecast ability of CGsNet and PhyDNet both degrades slightly. For strong wind (the threshold is at 13.9

m/s), CGsNet still has a certain forecasting ability, with an HSS of 0.20 (95% CI: 0.07; 0.24) and a POD of 0.22 (95% CI: 0.10; 0.31), which is superior to the performance of PhyDNet. This means that the developed attention module can effectively enhance modeling ability of CGsNet, making CGsNet a valuable tool for nowcasting ASWS during CGs events, particularly for strong gusts that are typically challenging to forecast using traditional methods.

**Table 2.** Quantitative results of CGsNet and PhyDNet on ASWS nowcasting. 95% CI represent the 95% confidence intervals of the indices.

| Model | ASWS (m/s) | CSI 95% CI | HSS 95% CI | POD 95% CI | MAE 95% CI | RMSE 95% CI |
|---|---|---|---|---|---|---|
| CGsNet | 8.0 | 0.41 (0.33; 0.49) | 0.54 (0.47; 0.61) | 0.59 (0.51; 0.66) | 1.60 (1.46; 1.80) | 2.26 (2.00; 2.54) |
| | 10.8 | 0.31 (0.22; 0.40) | 0.42 (0.32; 0.50) | 0.47 (0.34; 0.60) | 2.19 (1.93; 2.51) | 3.07 (2.62; 3.56) |
| | 13.9 | 0.15 (0.04; 0.21) | 0.20 (0.07; 0.24) | 0.22 (0.10; 0.31) | 2.90 (2.26; 2.66) | 4.22 (3.25; 5.10) |
| PhyDNet | 8.0 | 0.39 (0.31; 0.47) | 0.52 (0.44; 0.59) | 0.55 (0.46; 0.62) | 1.71 (1.55; 1.91) | 2.40 (2.14; 2.68) |
| | 10.8 | 0.28 (0.20; 0.38) | 0.38 (0.28; 0.47) | 0.41 (0.29; 0.53) | 2.40 (2.12; 2.76) | 3.33 (2.90; 3.83) |
| | 13.9 | 0.12 (0.03; 0.19) | 0.16 (0.05; 0.21) | 0.19 (0.09; 0.28) | 3.10 (2.39; 3.97) | 4.54 (3.59; 5.47) |

We further investigated the performance of models by drawing the CSI, HSS, MAE and RMSE curves at all nowcasting lead time stamps. As shown in Figure 4, in the cases with ASWS > 8.0 m/s, the prediction quality of both CGsNet and PhyDNet gradually decreases as the forecasting time increases. The CSI and HSS of CGsNet always retain top positions at any time. Specifically, for CGsNet, the value of CSI/HSS drops from approximately 0.56/0.70 (lead time at 6 min) to 0.40/0.45 (lead time at 120 min) in the nowcasting results, and the RMSE (MAE) increases from approximately 2.5 (1.7) m/s for the first lead time to 5.1 (3.7) m/s for the last lead time. The case of ASWS > 10.8 m/s is similar to that of ASWS > 8.0 m/s; the prediction quality of both models gradually decreases as the lead time increases, but the performances are degraded compared to that with a threshold of 8.0 m/s.

In addition, the evaluation curves are depicted with respect to different nowcasting lead times at the 13.9 m/s threshold Figure 4. The results are similar to the ASWS at the other two thresholds. It is also clear that CGsNet always has the performance on CSI and HSS for all steps. In the first 12 minutes, the performance of CGsNet decreases rapidly and then gradually decreases slowly. For PhyDNet, there is a slow decline in both CSI and HSS during the 0-2 hour forecasting period. It is worth noting that the nowcasting performance of PhyDNet is close to CGsNet after 108 min, but on the whole, CGsNet with the additional attention mechanisms is better than that of PhyDNet. Although the performance of CGsNet is worse than that with the other two thresholds, it still exhibits forecasting skill within 2 h. Overall, the above results confirm the effectiveness of the developed attention mechanism, which is potential for nowcasting ASWS in CGs events.

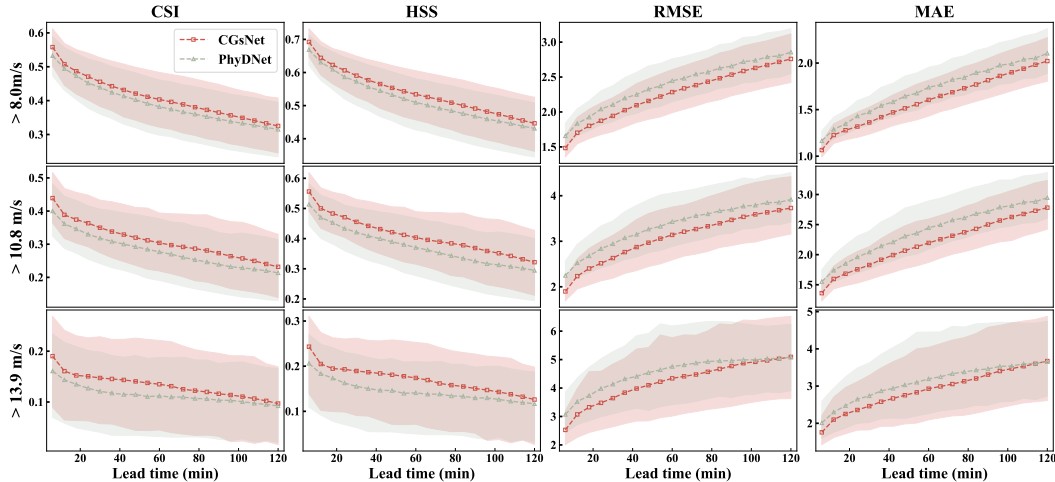

**Figure 4.** The CGsNet and PhyDNet results for different nowcasting lead times of ASWS at thresholds of 8.0 m/s, 10.8 m/s, and 13.9 m/s.The shaded pink and green areas represent the 95% confidence intervals of the CGsNet and PhyDNet indices, respectively.

## 4.2 Case study of ASWS nowcasting

To further prove the usefulness of CGsNet in detail, we show examples in two meteorologically important weather cases produced by CGsNet and PhyDNet. The first case occurred on April 30, 2021, when eastern China was affected by cold air due to the influence of the northeast China cold vortex. This cold air accompanied by the cold vortex moved eastward and southward, intersecting with some warm and humid air currents in eastern China. Then, the areas along the Yangtze River and north of Jiangsu suffered from SCW, including sudden CGs and hail. Because of the small scale and rapid development of

these CGs, numerical models usually cannot forecast this process accurately.

Specifically, hail and large-scale thunderstorms occurred from 18:00 to 22:00 (BJT) in parts of Nantong, Jiangsu, with a maximum wind speed of up to 47.9 m/s in some areas. Figure 5 shows the in situ observations and forecasted ASWS during this period. The CGs developed strongly at 20:12, and the ASWS exceeded 17.2 m/s in some areas. Then, the ASWS gradually decreased after 20:12 for 120 minutes. It is clear that the CGsNet model has better forecasting skill than PhyDNet (Figure 5).

CGsNet accurately forecasts the weakening trend of the CGs, while PhyDNet tends to overestimate the weakening trend of CGs. Additionally, the observed strong gusts continuously moved to the southeast as the lead time continued. CGsNet forecasts this phenomenon precisely, although in the later stage, the locations of the forecasted strong gusts are not exactly the same as in the observations. Specifically, the CGsNet model struggles to accurately forecast regions with strong gusts (ASWS > 10.8 m/s) and produced nowcasts that are slightly northward (approximately 50 km) of the observed locations. Additionally, in regions

with strong gusts (ASWS > 10.8 m/s), both CGsNet and PhyDNet underestimate the ASWS values, with PhyDNet showing a larger underestimation. Some false reports are also found from both models in the areas where 8.0 m/s < ASWS < 10.8 m/s, indicating that the modeling ability of CGsNet is limited.

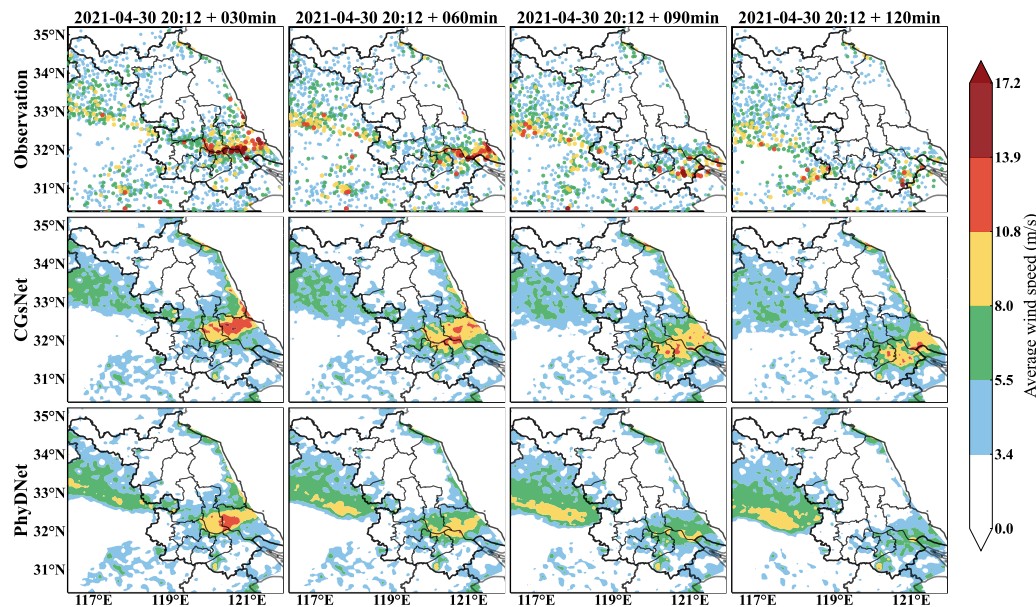

**Figure 5.** Observations (first row), CGsNet forecasts (second row), and PhyDNet forecasts (third row) of ASWS in eastern China, 30 April 2021, 20:12-22:12 BJT. Note that forecasting started at 20:12, and the observations and forecasts are shown at intervals of 30 min.

The second case is also typical convection weather. It was influenced by the low-level shear lines and strengthened by southwest warm and humid airflow on 15 May 2021. This process occurred in central and eastern Jiangsu, accompanied by heavy precipitation and strong gusts, causing serious economic losses. The observations and forecasts of ASWS for this case are presented in Figure 6.

In the second case, CGs are produced with the development of the squall line in the Jiangsu area. The strong gust position is linearly distributed, moving eastward from western Jiangsu to central Jiangsu between 13:00 and 15:00 (BJT). Figure 6 clearly shows that in most of the forecasting period, CGsNet outperforms PhyDNet due to its improved modeling capability. The results demonstrate that CGsNet can accurately forecast the ASWS values in most areas, although the strong gust values are slightly smaller than the observations in some areas. For PhyDNet, it shows a more pronounced underestimation of ASWS values compared to CGsNet. Furthermore, it is worth noting that in the later stages of prediction (e.g., lead time at 120 min), the forecasted CGs positions by PhyDNet are closer to the observations. Similar results can also be found in the RMOS forecasts (Figure S1 and S2 in the Supplement), which are the other output variables of CGsNet. Overall, the results further confirm that CGsNet is effective and accurate for ASWS nowcasting. Based on the reliable ASWS forecasts, nowcasting on the PWGS of CGs may be conducted.

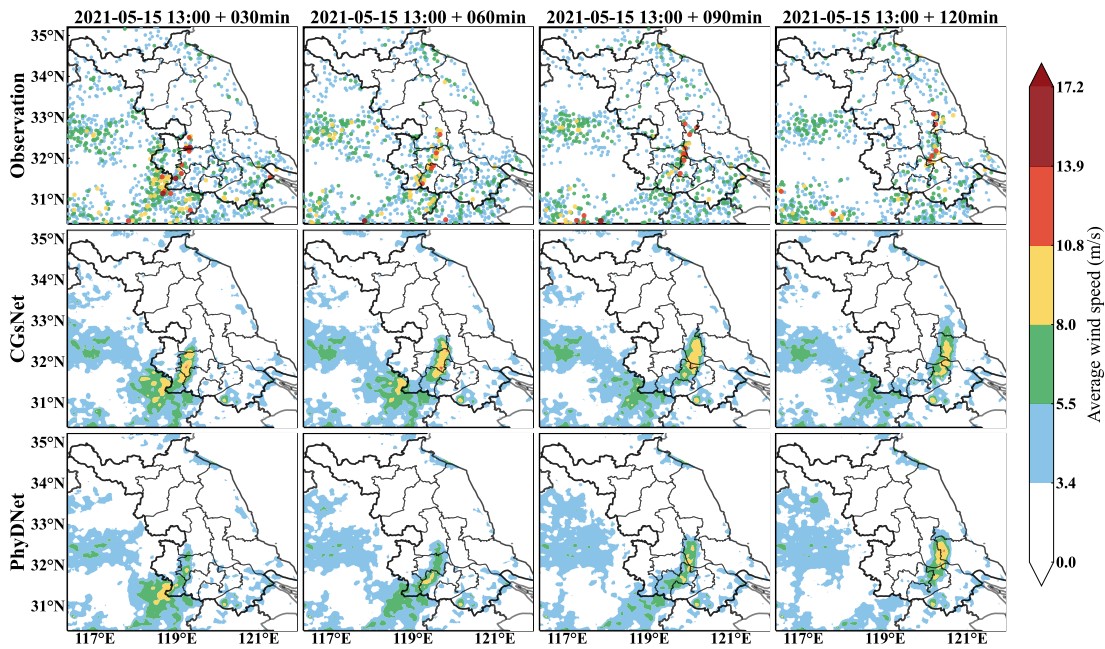

**Figure 6.** Same as Figure 5, but for 15 May 2021, 13:00-15:00 BJT.

## 4.3 Peak wind gusts speed nowcasting

ASWS reflects the average speed of CGs, which is not sufficient to determine the actual impact of CGs. The maximum momentary wind speed (e.g., PWGS) can reflect the actual peaks of CGs and is the key factor for the assessment and analysis of weather hazards. Thus, this section focuses on evaluating the reliability and accuracy of CGsNet on nowcasting PWGS.

### 4.3.1 Gust factor

The gust factor (GF) is the ratio of the wind gust to the wind speed (Harris and Kahl, 2017), and it plays an important role in wind conversion, the practice of converting between average wind speeds over different time periods (Harper et al., 2010). Kahl et al. (2021) showed that gust factors associated with strong winds (>7.65 m/s) were approximately 1.5 in the NW and NNW sectors of Rapid City, South Dakota, and ranged from 1.61 to 2.31 on different directionals at Waukegan. This directional variation in the GF was largely due to the site-specific heterogeneity in surface roughness (Suomi et al., 2013). Harris and Kahl (2017) investigated the GF in Milwaukee, Wisconsin, using Automated Surface Observation System (ASOS) wind measurements from 2007 to 2014 and calculated a mean GF of 1.74. Similarly, we determined the GF using the measured

wind data, i.e., ASWS and PWGS observations. Here, we do not consider the direction due to the relatively flat terrain in eastern China; instead, the average GF is explored.

We take the observed maximum ASWS within an hour as the hourly maximum ASWS to match the temporal resolution of the PWGS because the observed PWGS was reported for each hour at all sites in eastern China. Then, the GF is calculated as:

$$GF = \frac{PWGS}{hourly\ maximum\ ASWS} \tag{13}$$

According to the above formula, the observed wind data from April to September 2021, including 32015 stations data, were used to calculate the GF. The corresponding scatter plots of the hourly maximum ASWS and PWGS can be found in the Supplement (Figure S3). As a result, we found a GF of 1.77.

### 4.3.2 Model performance on PWGS nowcasting

Utilizing the calculated GF, we first estimated the 0-2 h PWGS nowcasts by multiplying the GF by the ASWS nowcasts from CGsNet. Then, the PWGS nowcasts were compared with INCA results to demonstrate the accuracy of CGsNet model and confirm the justifiability of the GF. Note that the results are in one-hour forecasts because the in situ PWGS observations and INCA results are reported for each hour.

The overall evaluation results, along with 95% confidence intervals, of CGsNet and INCA on PWGS nowcasting are given in Table 3. The results reveal that CGsNet significantly outperforms INCA on our testing dataset, and it achieved outstanding performance of all criteria. Moreover, as the threshold increases, the superiority of CGsNet becomes more and more obvious, e.g, for the PWGS threshold of 17.2 m/s, CGsNet still keep a decent modeling capability (CSI =0.15 (95% CI: 0.09; 0.19), POD=0.25 (95% CI: 0.15; 0.34)), which is much higher than INCA (CSI =0.03 (95% CI: 0.01; 0.05), POD=0.07 (95% CI: 0.03; 0.12)).

**Table 3.** Quantitative results of CGsNet and INCA on PWGS nowcasting with 95% confidence intervals (in brackets). Note that the values in the first row of each metric represent CGsNet results, while the second row is INCA results.

| Evaluation Metrics | 10.8 m/s | 13.9 m/s | 17.2 m/s | 20.8 m/s |
|---|---|---|---|---|
| CSI | 0.27 (0.25; 0.28) | 0.22 (0.18; 0.26) | 0.15 (0.09; 0.19) | 0.06 (0.02; 0.08) |
| | 0.11 (0.09; 0.12) | 0.06 (0.03; 0.08) | 0.03 (0.01; 0.05) | 0.02 (0.00; 0.03) |
| POD | 0.35 (0.32; 0.36) | 0.31 (0.26; 0.37) | 0.25 (0.15; 0.34) | 0.12 (0.04; 0.18) |
| | 0.30 (0.28; 0.32) | 0.13 (0.09; 0.18) | 0.07 (0.03; 0.12) | 0.04 (0.01; 0.07) |
| BIAS | 0.63 (0.61; 0.65) | 0.73 (0.68; 0.78) | 0.94 (0.76; 1.13) | 1.21 (0.96; 1.54) |
| | 2.07 (1.88; 2.42) | 1.51 (1.44; 1.68) | 1.27 (1.21; 1.37) | 1.14 ( 1.07; 1.30) |
| FAR | 0.45 (0.44; 0.46) | 0.57 (0.52; 0.62) | 0.73 (0.69; 0.80) | 0.90 (0.87; 0.96) |
| | 0.85 (0.83; 0.88) | 0.91 (0.87; 0.95) | 0.94 (0.91; 0.99) | 0.96 (0.94; 1.00) |

For a more detailed comparison, the PWGS results from CGsNet and INCA for 0-2 h nowcasting lead times at different thresholds are shown in Table 4. At the lead time of 0-1 h, the CSI and POD of CGsNet always keep top positions at different thresholds. The FAR of CGsNet is also lower than that of INCA. Meanwhile, except for the threshold of 20.8 m/s, CGsNet consistently exhibits better BIAS than INCA. Additionally, the quality of CGsNet and INCA both degraded with the forecasting time increasing. Specifically, the performance of CGsNet still keep a forecasting skill at the lead time of 1-2 h, while INCA is similar to the first hour, it is lack of skill, the CSI at all thresholds is lower than 0.1. Compared to other thresholds, INCA's performance is closest to CGsNet's when the threshold is set to 20.8m/s. Additionally, it's worth noting that the POD of INCA at the threshold of 10.8 m/s is a little larger than the CGsNet (0.29 (95% CI: 0.21; 0.37) vs. 0.28 (95% CI: 0.16; 0.41)) at the lead time of 1-2 h. This is due to the high false alarm of INCA, thus, it has the high FAR and BIAS.

**Table 4.** The PWGS evaluation results from CGsNet and INCA for different nowcasting lead times at thresholds of 10.8 m/s, 13.9 m/s, 17.2 m/s and 20.8 m/s. Note that the values in the brackets represent 95% confidence intervals, and values in the first row of each metric represent CGsNet results and the second row is INCA results.

| Lead time of 0-1 h | | | | |
|---|---|---|---|---|
| **Evaluation Metrics** | **10.8 m/s** | **13.9 m/s** | **17.2 m/s** | **20.8 m/s** |
| CSI | 0.31 (0.23; 0.40) | 0.26 (0.16; 0.34) | 0.17 (0.10; 0.24) | 0.06 (0.02; 0.09) |
| | 0.13 (0.10; 0.16) | 0.06 (0.03; 0.09) | 0.04 (0.01; 0.06) | 0.02 (0.01; 0.04) |
| POD | 0.40 (0.28; 0.52) | 0.37 (0.24; 0.49) | 0.28 (0.15; 0.41) | 0.14 (0.05; 0.26) |
| | 0.31 (0.23; 0.37) | 0.13 (0.06; 0.19) | 0.07 (0.02; 0.13) | 0.03 (0.01; 0.07) |
| BIAS | 0.69 (0.50; 0.89) | 0.79 (0.56; 0.99) | 0.99 (0.59; 1.36) | 1.43 (0.69; 2.36) |
| | 1.74 (1.37; 2.25) | 1.30 (0.89; 1.78) | 1.10 (0.56; 1.83) | 1.00 (0.33; 1.93) |
| FAR | 0.42 (0.34; 0.52) | 0.53 (0.45; 0.64) | 0.71 (0.63; 0.83) | 0.90 (0.86; 0.96) |
| | 0.82 (0.78; 0.87) | 0.90 (0.83; 0.96) | 0.93 (0.86; 0.98) | 0.97 (0.94; 1.00) |
| Lead time of 1-2 h | | | | |
| **Evaluation Metrics** | **10.8 m/s** | **13.9 m/s** | **17.2 m/s** | **20.8 m/s** |
| CSI | 0.22 (0.13; 0.31) | 0.18 (0.09; 0.26) | 0.13 (0.04; 0.20) | 0.05 (0.01; 0.09) |
| | 0.09 (0.06; 0.12) | 0.05 (0.02; 0.08) | 0.03 (0.00; 0.06) | 0.02 (0.00; 0.04) |
| POD | 0.28 (0.16; 0.41) | 0.26 (0.12; 0.39) | 0.21 (0.06; 0.36) | 0.09 (0.01; 0.19) |
| | 0.29 (0.21; 0.37) | 0.14 (0.06; 0.22) | 0.07 (0.01; 0.12) | 0.05 (0.00; 0.10) |
| BIAS | 0.56 (0.35; 0.77) | 0.66 (0.41; 0.89) | 0.88 (0.47; 1.26) | 0.98 (0.36; 1.72) |
| | 2.43 (1.82; 3.36) | 1.73 (1.18; 2.46) | 1.45 (0.81; 2.23) | 1.29 (0.60; 2.20) |
| FAR | 0.49 (0.39; 0.61) | 0.61 (0.52; 0.75) | 0.76 (0.66; 0.90) | 0.91 (0.83; 1.00) |
| | 0.88 (0.84; 0.92) | 0.92 (0.86; 0.97) | 0.95 (0.87; 1.00) | 0.96 (0.87; 1.00) |

The PWGS forecasts for different hours of a day and months of a year are further calculated to evaluate the performance of CGsNet. The evaluation results for CSI are presented here. The results for POD, BIAS, and FAR can be found in the supplement

(Figure S4-S9). Figure 7 displays the CSI results of PWGS forecasts from CGsNet and INCA at different hours of a day. The PWGS samples were mainly obtained during the afternoon and night periods, as CGs events tend to occur during these times, particularly in the late afternoon and evening (Firouzabadi et al., 2019). The results demonstrate that CGsNet outperforms INCA in forecasting PWGS at different thresholds, with overall superior forecast performance. However, the performance of both models declines as the PWGS threshold increases. Despite CGsNet's superiority, its performance is less stable than INCA across different hours, exhibiting significant variability. For example, at 21:00 (BJT), the performance of CGsNet declines, and even when PWGS > 10.8 m/s, its performance (CSI and POD) is worse than INCA. At PWGS > 17.2m/s, the confidence intervals of CGsNet and INCA are both wide, while some CSI and POD values of INCA fall outside the confidence intervals and exhibit almost no predictability for CGs during 20:00-24:00 (BJT) (CSI=0, POD=0). Additionally, when the PWGS threshold is 20.8 m/s, neither CGsNet nor INCA is skillful for CGs nowcasting between 20:00 and 24:00 (BJT), with CGsNet showing a higher FAR than INCA (Figure S6). These may be attributed to the fact that strong gusts occur less frequently during these times, resulting in fewer high-value PWGS samples and highlighting the imbalance of wind data.

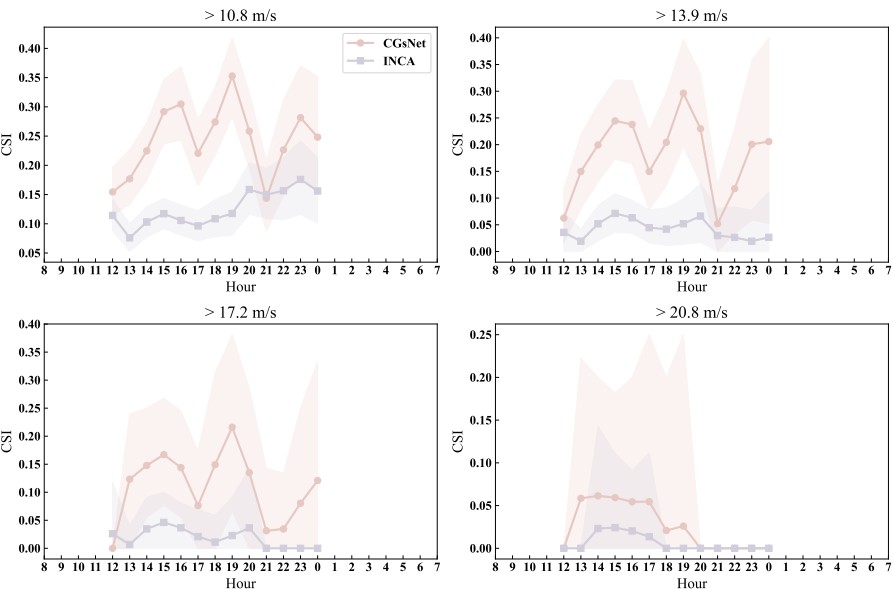

**Figure 7.** The CSI results of PWGS forecasts from CGsNet and INCA for different hours of a day at thresholds of 10.8 m/s, 13.9 m/s 17.2 m/s, and 20.8 m/s.The shaded pink and purple areas represent the 95% confidence intervals of the CGsNet and INCA indices, respectively.

The CSI results for the PWGS forecasts of CGsNet and INCA in different months of a year are illustrated in Figure 8. At different PWGS thresholds, CGsNet outperform INCA in terms of CSI and POD for PWGS forecasts from May to July, while FAR is lower than INCA. However, there are large confidence intervals in the calculated evaluation metrics at PWGS thresholds of 17.2 m/s and 20.8 m/s due to the low number of strong gusts samples, leading to uncertainty. Moreover, for PWGS > 20.8m/s, both CGsNet and INCA exhibit almost no predictability for May and June PWGS forecasts. Additionally,

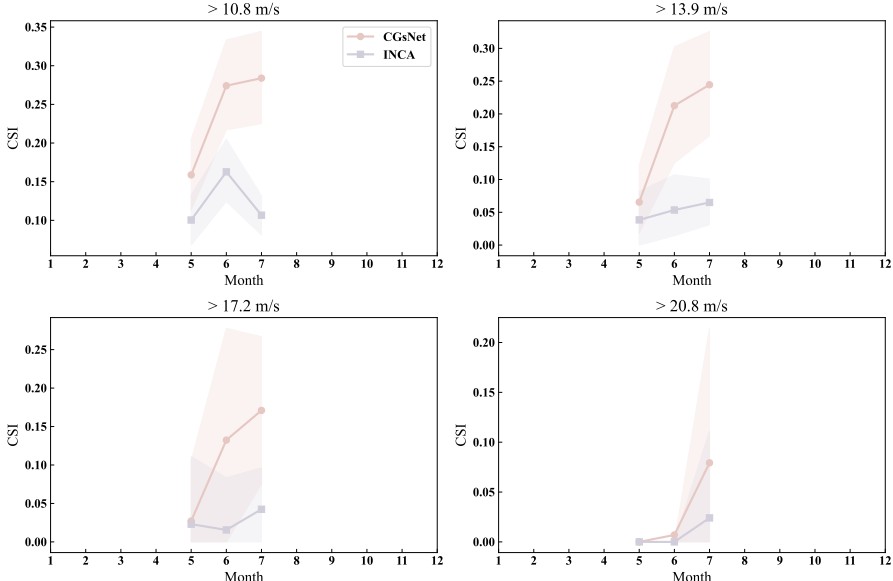

**Figure 8.** The CSI results of PWGS forecasts from CGsNet and INCA for different months of a year at thresholds of 10.8 m/s, 13.9 m/s 17.2 m/s, and 20.8 m/s.The shaded pink and purple areas represent the 95% confidence intervals of the CGsNet and INCA indices, respectively.

CGsNet's forecast performance generally improves with increasing months, while INCA exhibits a fluctuation in June PWGS forecasting and overall better performance in July PWGS forecasting compared to May PWGS forecasting.

To better understand the performance of the model, we evaluated the forecasting performance of CGsNet and INCA for PWGS using various metrics across different geographic locations within the study area. Figure 9 displays the results for
CSI, while the results for POD, BIAS, and FAR are provided in Figure S10-S12 in the supplement. CGsNet shows better performance than INCA in forecasting PWGS at different thresholds. As the threshold increases, the forecasting performance of CGsNet decreases. This may be due to two reasons. Firstly, there is a reduction in the number of observed PWGS samples, as some areas do not experience CGs events, resulting in an evaluation value of 0 for those areas. Secondly, CGsNet has limitations in its forecasting ability, particularly for extreme PWGS (>20.8 m/s), which is mainly owing to the imbalance of
observed strong and weak wind data. Specifically, for the PWGS thresholds are at 10.8 m/s and 13.9 m/s, CGsNet exhibits a outstanding ability to forecast PWGS for diverse regions in Jiangsu (excluding the edge areas), with most areas achieving CSI and POD values above 0.8. Conversely, INCA's forecasting performance is poor for most areas, with relatively better results in the southwest of Jiangsu. However, even in these regions, INCA's performance is still inferior to that of CGsNet. Although INCA outperforms CGsNet in terms of CSI and POD evaluation results for a PWGS threshold of 10.8 m/s in the Anhui Province
region, but INCA exhibits high FAR and poor BIAS in this area. For PWGS > 17.2 m/s, CGsNet shows good forecasting results in central, northern, and southern Jiangsu, while INCA only performs well in a few stations in southwestern Jiangsu. When PWGS > 20.8 m/s, both CGsNet and INCA exhibit poor forecasting skill, and only a few stations can be effectively forecasted.

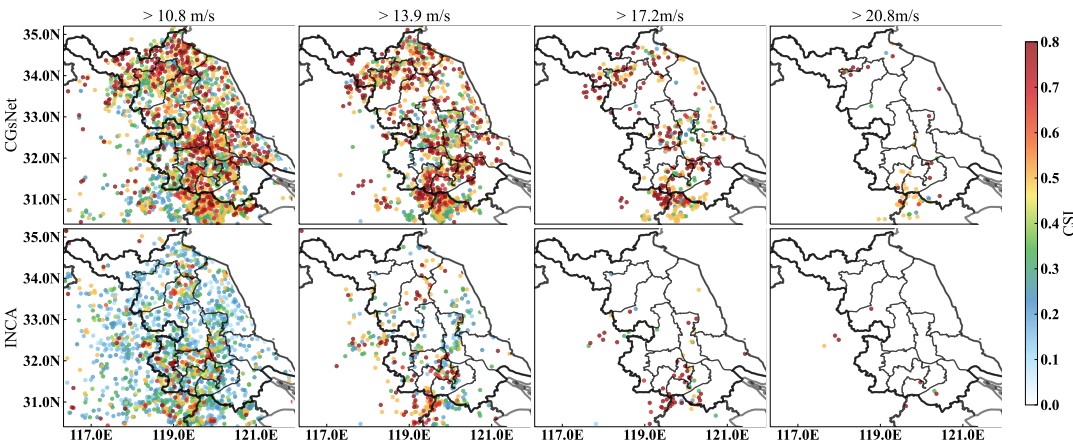

**Figure 9.** The CSI results of PWGS forecasts from CGsNet and INCA for different areas at thresholds of 10.8 m/s, 13.9 m/s 17.2 m/s, and 20.8 m/s.

Additionally, CGsNet and INCA show poor forecasting performance in regions outside of Jiangsu, due to the dominance of CGs occurring in Jiangsu in the PWGS test dataset, with limited samples from outside the Jiangsu. Obtaining more PWGS evaluation samples outside of Jiangsu in future studies could address this issue.

A scatter plot of observed and forecasted PWGS is presented in Figure 10 for further comparison. The results indicate

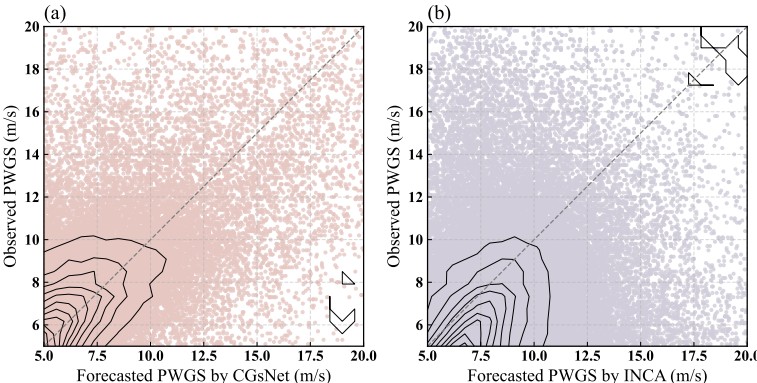

**Figure 10.** Scatter plot of observation and forecasted PWGS, (a) CGsNet forecasts vs. observation, (b) INCA forecasts vs. observation. The density distribution curves are represented by the black lines.

that although CGsNet slightly underestimates PWGS, it performs well for PWGS values less than about 12 m/s. However, its performance decreases, and a bias in PWGS forecasts is observed for PWGS values greater than approximately 12 m/s. In contrast, INCA has a obvious overestimation for PWGS < 12 m/s, and the PWGS forecasts have a large deviation, not

corresponding well to the observations. Additionally, the performance of INCA for high PWGS values is also poor, which is of great concern. In summary, the results indicate that the developed CGsNet is helpful to improve the accuracy of CGs nowcasting and more skillful than INCA, although it tends to underestimate strong gusts.

### 4.3.3 Case study of PWGS nowcasting

To illustrate the perfromance of CGsNet more intuitively, the INCA forecasts in two instances are presented for comparison. The first case is on June 23, 2022, short-term heavy precipitation and CGs occurred over a large area in Jiangsu Province because of the influence of the northeast China cold vortex. As shown in Figure 11, for the first nowcasting hour (19:00 BJT), the CGs were mainly located in central Jiangsu, and the PWGS at some sites exceeded 20.8 m/s. Then, the CGs moved to the southeast of Jiangsu from 19:00 to 20:00 (BJT), occurring in Nantong and surrounding areas, and the PWGS gradually decreased. During this process, it is clear that CGsNet can effectively forecast the location of the CGs, especially for the first hour of nowcasting (19:00 BJT). However, CGsNet underestimates the PWGS at some sites. INCA does not accurately forecast the occurrence of CGs at 19:00 (BJT); the area in which PWGS > 17.2 m/s is too small, and the PWGS is also significantly underestimated. Besides, INCA failed to forecast the occurrence of CGs at 20:00 (BJT), and the forecasted PWGS values were all below 17.2 m/s.

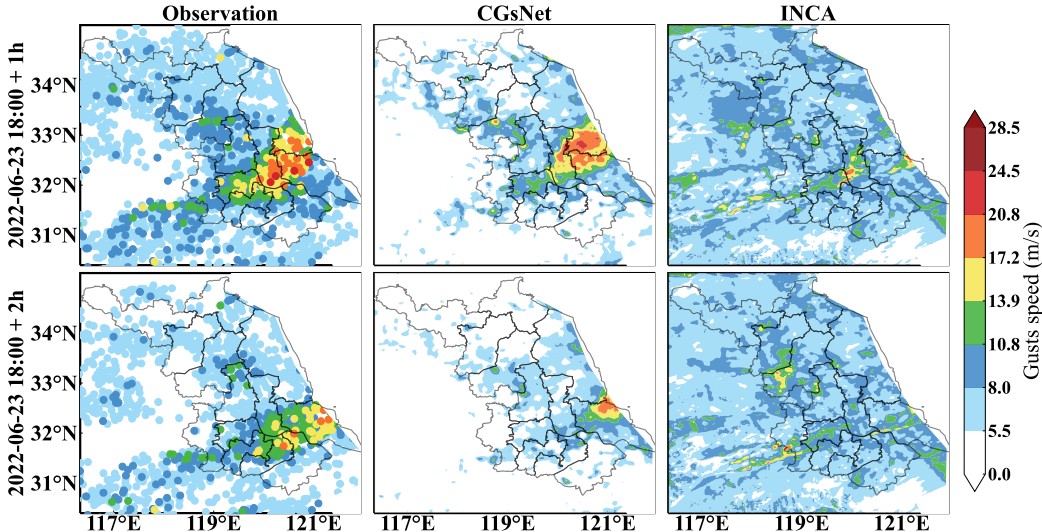

**Figure 11.** Observations and forecasts of PWGS in eastern China on June 23, 2022, 18:00-20:00 BJT. Note that the forecasting started at 18:00. The first column shows the observations at 19:00 BJT and 20:00 BJT. The last two columns show the PWGS forecasts obtained by CGsNet and INCA.

The performance of CGsNet and INCA at different PWGS thresholds for the case on on June 23, 2022 are shown in Figure 12. In particular, the PWGS at thresholds of 8.0 m/s, 10.8 m/s, 13.9 m/s and 17.2 m/s is compared because the PWGS is below 20.8 m/s over most of the study area at 20:00 (BJT). The performance of CGsNet and INCA both decrease as the threshold

of PWGS increases, but CGsNet significantly outperforms INCA when PWGS at thresholds of 8.0 m/s, 10.8 m/s, 13.9 m/s. For example, in the cases of PWGS > 10.8 m/s, CGsNet achieves outstanding performance on the criteria, with a CSI of 0.34 (95% CI: 0.27; 0.43) and a POD of 0.43 (95% CI: 0.34; 0.53), which are better than the corresponding INCA values (CSI=0.15 (95% CI: 0.10; 0.21), POD=0.26 (95% CI: 0.18; 0.35)). It is noteworthy that the POD (0.49 (95% CI: 0.44; 0.55)) of CGsNet is inferior to that of INCA (0.64 (95% CI: 0.59; 0.70)) at the 8.0 m/s threshold. This may be because CGsNet has missed forecasting some PWGS > 8.0 m/s (as shown in the east of Jiangsu Province in Figure 11), while INCA has falsely detected the PWGS in areas where it is actually less than 8.0 m/s. INCA predicted large areas with PWGS > 8.0 m/s, despite PWGS < 8.0 m/s or the absence of strong gusts in many regions, leading to a high FAR and BIAS. Besides, the BIAS of CGsNet (0.70 (95% CI: 0.55; 0.86)) is also worse than that of INCA (0.97 (95% CI: 0.76; 0.1.22)) at a threshold of 10.8 m/s, possibly because some areas with PWGS < 10.8 m/s are mispredicted by INCA as PWGS > 10.8 m/s. In this condition, the BIAS of INCA is nearly optimal, but the performance on FAR and POD is poor. Again, the advantages of CGsNet become increasingly obvious with increasing PWGS threshold (13.9 m/s and 17.2 m/s). In particular, for PWGS > 17.2 m/s, INCA is nearly unable to perform CGs nowcasting, while CGsNet still shows nowcasting skill, but the forecast uncertainties of both models are relatively large. This is because the samples for PWGS observations at this threshold are small, leading to wide confidence intervals calculated by bootstrapping method.

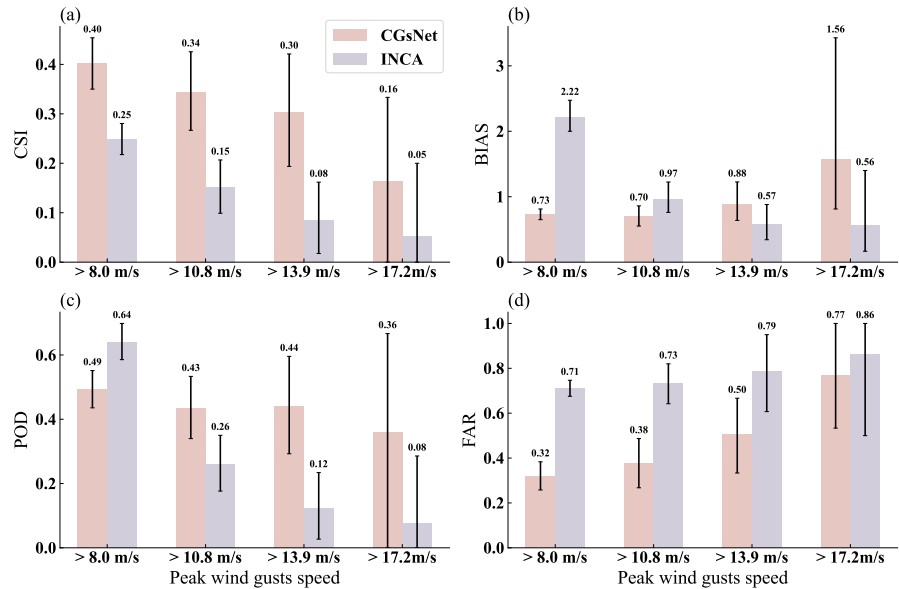

**Figure 12.** Comparison results of CGsNet and INCA on PWGS at thresholds of 10.8 m/s, 13.9 m/s, and 17.2 m/s on June 23, 2022, 18:00-20:00 BJT (Black error bars represent 95% confidence intervals).

The other CGs case on 26 July 2022 was also selected to further verify the nowcasting ability of the CGsNet model. For this case, with the movement and development derecho in eastern China, Jiangsu and northern Zhejiang were hit by SCW (i.e., CGs, short-term heavy precipitation, and tornadoes). Figure 13 shows the observed and forecasted PWGS. Specifically, central and southern Jiangsu suffered from CGs during 14:00-16:00 (BJT), and there was PWGS > 28.5 m/s in some areas. From

15:00 to 16:00 (BJT), the CGs gradually moved eastward with slight weakening. Compared with the observations, CGsNet was shown to be accurate in capturing the location of the CGs, despite underestimating the PWGS, which was greater than 20.8 m/s. INCA did not accurately forecast the area where the CGs would occur and underestimated the PWGS. In addition, neither CGsNet nor INCA accurately forecasted the CGs that occurred in northern Zhejiang at 15:00 and 16:00 (BJT); the CGsNet forecasts were weaker than the observations, while the INCA forecasts were in the wrong area. The results demonstrate that CGsNet is better than INCA and shows excellent performance in CGs nowcasting.

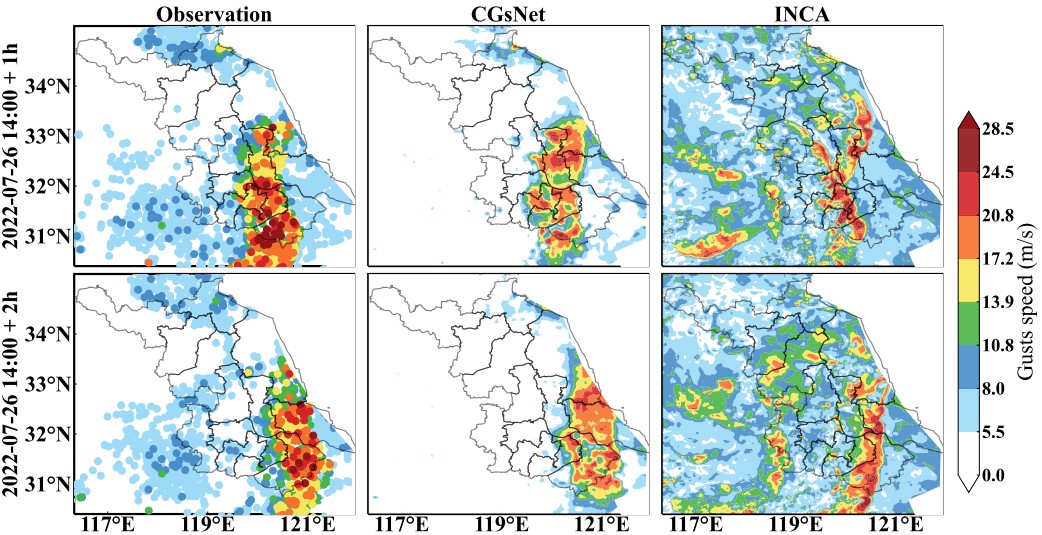

**Figure 13.** Same as Fig. 11, but for July 26, 2022, 14:00-16:00 BJT.

In addition, Figure 14 shows the performance of CGsNet and INCA at different PWGS thresholds. Unlike the first case, we evaluate the PWGS at a threshold of 20.8 m/s, since many sites have PWGS > 20.8 m/s in this case. The results indicate that the CSI and POD of both methods show a general decrease with increasing thresholds, and the BIAS and FAR increase. The CGsNet forecasts achieve better performance than those of INCA at all thresholds; e.g., the evaluated POD reaches 0.51 (95% CI: 0.38; 0.64) when the threshold is 17.2 m/s, which is 147.62% higher than that of INCA. Additionally, INCA has a poor performance on CSI and FAR at each PWGS threshold, which is due to the large location deviation of the INCA forecasts with respect to where CGs occurred. It is worth noting that when the threshold of PWGS is 20.8 m/s, the uncertainties of CSI and POD are both high, similar to the situation when the threshold of PWGS is 17.2 m/s in the first case. Overall, the results further highlight the usefulness of the proposed CGsNet in CGs nowcasting. It also demonstrates the effectiveness of the GF.

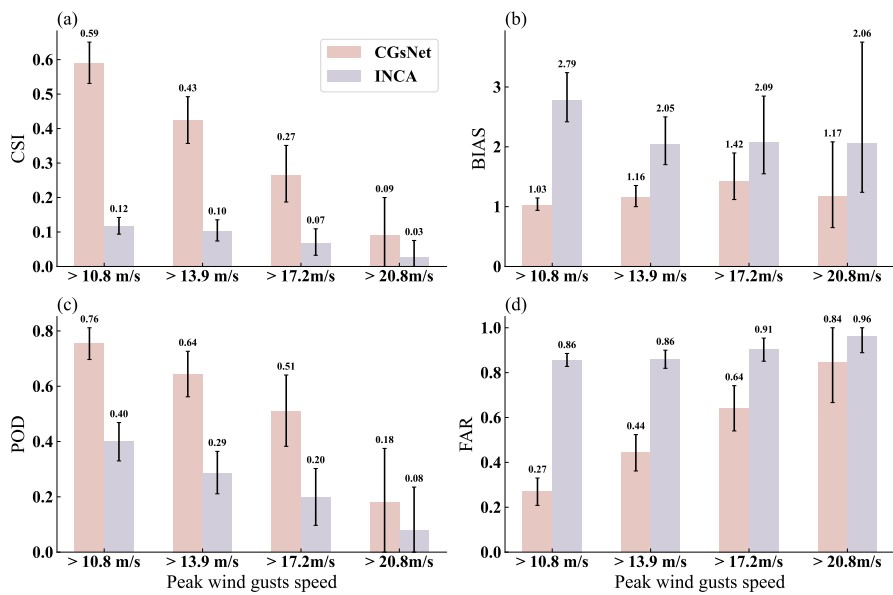

**Figure 14.** Same as Fig. 12, but for July 26, 2022, 14:00-16:00 BJT.

## 5 Discussion and conclusions

In this study, a physics-constrained deep learning model was proposed for CGs nowcasting. This model was established based on the extended PhyDNet and is called CGsNet. Then, by using observed ASWS and RMOS datasets, CGsNet was trained for CGs nowcasting in eastern China. CGsNet can produce ASWS nowcasts of 6-120 min with a temporal resolution of 6 min

and a spatial resolution of 1 km. To estimate the PWGS forecasts, we multiplied the forecasted ASWS by the GF, which was determined from the ratio of the historical observed ASWS to the PWGS. The PWGS forecasts had the same spatiotemporal resolution as the ASWS forecasts. This is the first time that minute-level CGs nowcasting has been achieved for eastern China.

Several metrics for evaluating the performance of CGsNet were analyzed, confirming the effectiveness of the CGsNet model for ASWS and PWGS nowcasting. The results of our comparison between the ASWS forecasts generated by CGsNet and the

baseline (PhyDNet) revealed the effectiveness of the newly developed attention module. The PWGS forecasts were also compared with the INCA results, and the results demonstrated that the performance of CGsNet is better than that of INCA. CGsNet can effectively forecast the location and evolution of CGs, and the PWGS forecasts are a good match with the observations. In contrast, INCA does not accurately determine the locations and strengths of CGs, with low POD and CSI and high BIAS and FAR. The superiority of the PWGS results also proves that the calculated GF is reliable.

In addition to these achievements, there are some points requiring further discussion and investigation. For example, the intensity of ASWS and PWGS sometimes is underestimated (especially for strong gusts, i.e., PWGS>20.8 m/s) or there is sometimes a spatial offset between the forecasted and observed strong gusts, which may be caused by several different factors. For ASWS forecasts, this may be due to the limited modeling ability of CGsNet; that is, the model has not fully learned

the nonlinear variation characteristics of CGs, which mainly due to the scarcity of strong gusts data. In addition, the input
variables may be insufficient, and it is difficult to accurately forecast small-scale and nonstationary CGs with only radar data
and observed wind data. For PWGS forecasts, the error from ASWS and GF, and thus their combination, leads to the deviation
of the predicted PWGS.

Therefore, in future work, more meteorological factors related to CGs, such as the three-hour pressure change and relative
humidity, should be considered in CGs nowcasting. And the imbalance of samples in the dataset can be further improved.
The predictive ability of the model can then be improved to some extent by reforming the training dataset. In addition, a
more accurate GF should be calculated by using years of wind observations or computing site-specific/direction-specific GFs.
Although our experiments were performed in eastern China, the proposed CGs nowcasting technology can be generalized
to a wide range of areas. Besides, CGs nowcasting technology could also be applied to forecast other phenomena, such as
convection or hail, for which real-time-response models are crucially needed.

*Code availability.* The source code and the used data are available at this repository https://doi.org/10.7910/DVN/PIZU7V.

*Author contributions.* Haixia Xiao was responsible for writing the original draft and completing the result analysis. Yaqiang Wang directed
the study and wrote and reviewed the paper. Yu Zheng was responsible for developing the CGsNet model and visualizing the results.
Yuanyuan Zheng provided useful comments on this study. The INCA and AWS datasets were processed by Xiaoran Zhuang. The RMOS
data were acquired and processed by Hongyan Wang and Mei Gao.

*Competing interests.* The authors declare that they have no conflicts of interest.

*Acknowledgements.* The authors thank the anonymous reviewers and editors for their insightful comments and suggestions, which have
signifcantly improved this paper. This work was supported by the National Natural Science Foundation of China, Grant U2142210; CAMS
project 2020Z011; the Basic Creative Research Fund of Huafeng Meteorological Media Group CY—J2020001; and the Key Innovation
Team of China Meteorological Administration (CMA2022ZD07).

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
