# Peer review of "Convective Gusts Nowcasting Based on Radar Reflectivity and a Deep Learning Algorithm"

_Geoscientific Model Development, 2022_

## Referee Comment (RC1)

Convective wind gusts (GCs) cause great structural damage and serious hazards. This paper designs a physics-constrained model (namely by an improved PhyDNet) for 0-2 h of quantitative CGs nowcasting with a spatial resolution of $0.01° \times 0.01°$ and a 6-minute temporal resolution. The structure of the forecast neural network is designed interesting, which contains a temporal attention module. In addition, this model combines sufficient situ observations and radar data. I admire the author's dedication to such detailed work. I only have some minor points to make before being published.

**I  DATA**

① The wind data the authors used is from automatic weather stations (AWSs), and they are few wind observation data on the seacoast (and even over the sea) from Figure 1b. I wonder how to interpolate the wind data over the sea, whether to set it to 0 directly. Please clarify the detailed information on the IDW interpolation method.

② If the wind data over the sea is 0, please give some analyses on whether this approach affects model performance, such as causing underestimations on PWGS to some extent.

③ The same interpolation problem also emerges in radar reflectivity. There are 10 weather radar stations in eastern China, and how to interpolate them into grided data. I suggest the authors exhibit some exemplificative image pairs for ASWS/radar data and processed gridded data.

④ Is the data the authors used open-source? If so, please clarify the web links of them.

**II  MODEL**

① Please clarify the detailed structures of the convolutional encoder and decoder in Appendix, and the feature map shapes of $h^E$, $h^p$, $h^c$, and $h^m$.

② What is the difference between $h^D$ and $h^m$? In Equation (4), $h^D$ is the summation of $h^p$ and $h^c$. However, in Figure 3, $h^D$ is from some transformation of $h^m$, and $h^m$ is the summation of $h^p$ and $h^c$. Please unify the illustration.

③ From the example of Figure 3, I wonder if there are 4 individual attention modules corresponding to the output length. And the effect of the attention is to find the most significant historical information from the input sequence. For example, when predicting $t$, the attention module can assign weights for $\{h_{t-4}^m, h_{t-3}^m, h_{t-2}^m, h_{t-1}^m\}$, and when prediction $t+1$, the attention module can assign weights for $\{h_{t-4}^m, h_{t-3}^m, h_{t-2}^m, h_{t-1}^m\}$ as well, or for $\{h_{t-3}^m, h_{t-2}^m, h_{t-1}^m, h_t^m\}$.

④ There is no $k$ on the right side of Equation (1), how to obtain $s_{tk}$ $(\forall k \in [1, K])$?

⑤ Please give the detailed calculation process of PhyDNet and ConvLSTM in Appendix.

⑥ About the physics-constraint concept mentioned in Introduction and Model architecture (namely PhyDNet), I think the authors should cite the "inductive biases" from doi.org/10.1038/s42254-021-00314-5, which can be interpreted by designing specialized network architectures that implicitly embed prior knowledge and satisfy a set of given physical laws.

**III  EXPERIMENT**

① Please append an ablation study on the proposed attention module.

---

## Referee Comment (RC2)

This paper describes the use of a neural network for predicting convective wind gusts at lead times of 6-120 minutes. The neural network includes several architectural components that have recently been developed in the field of deep learning. These include pieces of the recurrent neural net (RNN) architecture, where the prediction at the $k^{th}$ lead time becomes an input (predictor) to the neural network to predict at the $(k + 1)^{th}$ lead time; the attention mechanism, where the neural network can automatically determine the most important steps in a time series of predictor values; and PhyCell, which incorporates partial differential equations (PDE). The neural network predicts convective wind gusts on a 1-km grid covering eastern China. The authors compare their results to a baseline model called INCA and demonstrate (albeit without significance-testing) that their neural network, which they call CGsNet, outperforms INCA. Overall, the quality of both the writing and science are very good. I have a handful of major comments, which are made as inline comments in the attached PDF but also summarized below. Minor comments are not summarized below and can be seen only in the attached PDF.

**Major comments**

1. The abstract contains several unclear/unjustified statements. See inline comments.

2. Most hyperparameter choices are not justified at all. Whether hyperparameter choices were made via experimentation or *a priori* reasoning, the justification should be included in the paper. If hyperparameters were chosen by experimentation, the experiments should be documented (although I'm okay with putting most of the details in the Appendix or Supplemental Material); if chosen by *a priori* reasoning, the *a priori* reasoning should be explained. For examples of unjustified hyperparameter choices, see lines 84-85, lines 150-152, and Table 1 (and inline comments at these places).

3. It seems like the authors used some kind of subsetting to create their dataset, but the subsetting methodology is unclear. See inline comment on lines 86-88.

4. It is unclear how the authors handle radar data at different heights. See inline comment on lines 96-97.

5. The target variables (what the CGsNet is predicting) eventually become clear, but quite late in the paper. The target variables should be clear from the beginning. See inline comment on line 107.

6. Section 3.1 needs to be written more clearly. See inline comments for specific points that are unclear.

7. I don't understand how the authors split their data. Specifically, (a) I don't understand how the training and validation data are split; (b) I don't understand why they have such an incredibly small testing dataset for wind gusts. See inline comments on lines 146-149.

8. It seems like the authors inappropriately used their testing data for model selection; only the validation data should be used for model selection. See inline comment on line 153.

9. The comparison of a custom loss function to data augmentation is highly erroneous. See inline comment on line 164.

10. The results contain no confidence intervals or significance-testing -- in other words, no measure of uncertainty.  I emphasize this comment the most, since results without uncertainty are nearly meaningless.  All results in Figure 4 and Tables 2-4 should be accompanied by at least confidence intervals, if not also significance tests.  See inline comments for details.

11. The model evaluation is lacking in detail.  The authors present evaluation metrics based on the full testing dataset (least granularity) and some individual case studies (most granularity) -- but nothing in between (intermediate scales of granularity).  The paper should include evaluation metrics as a function of time of day (e.g., by hour), time of year (e.g., by month), geographic location (e.g., maps with POD, CSI, FAR, etc. at each 1-by-1-km grid cell), and predicted wind speed (i.e., the reliability curve).  At the very least, I want to see reliability curves and some division by time (either time of day or time of year).  Ideally, I would like to see the evaluation metrics broken down in all 4 ways that I have listed.  Presenting evaluation metrics at varying levels of granularity is crucial for understanding a model, especially for understanding its strengths and weaknesses.

12. The authors' interpretation of Figure 5 does not seem to be justified.  See inline comments on Figure 5 itself and also lines 226-227.

[revised manuscript text omitted]

* * *
Anonymous: Besides the one unclear point in the caption, I *really* like this figure. It is very well done and informative. Kudos!

Anonymous: This part does not make sense. "The input and output tensors calculated by the conv and deconv units" = 4 things: input tensor for conv unit, output for conv unit, input for deconv unit, output for deconv. But you match this description with 2 variables (h_i^E and h_i^D), not 4 variables.

Anonymous: Which years are for training, and which are for validation?

Anonymous: Why *so* few samples for PWGS? This is an extremely small dataset compared to the others. The confidence intervals for your PWGS-based evaluation scores must be huge...

Anonymous: Why such a small batch size? Did you experiment with other batch sizes?

Anonymous: How did you choose these

[revised manuscript text omitted]

> Anonymous: This table should contain confidence intervals (created, e.g., by bootstrapping the testing samples). Without a measure of uncertainty, it is hard to understand these results.

We further investigated the performance of CGsNet by drawing the CSI, HSS, MAE and RMSE curves at all nowcasting lead time stamps. As shown in Figure 4, in the cases with ASWS > 8.0 m/s, the prediction quality gradually decreased as the forecasting time increased. Specifically, the value of CSI/HSS dropped from approximately 0.56/0.70 (lead time at 6 min) to 0.40/0.45 (lead time at 120 min) in the nowcasting results, and the RMSE (MAE) increased from approximately 2.5 (1.7) m/s for the first lead time to 5.1 (3.7) m/s for the last lead time. The case of ASWS > 10.8 m/s (Figure 4b and 4e) is similar to that of ASWS > 8.0 m/s; its prediction quality gradually decreases as the lead time increases, but the performance is degraded compared to that with a threshold of 8.0 m/s.

[Figure]

> Anonymous: This figure should also contain confidence intervals -- displayed, e.g., as error bars.

**Figure 4.** The CGsNet results for different nowcasting lead times of ASWS at thresholds of 8.0 m/s, 10.8 m/s, and 13.9 m/s.

[Figure]

In addition, the evaluation curves are depicted with respect to different nowcasting lead times at the 13.9 m/s threshold. The results are similar to the ASWS at the other two thresholds. In the first 12 minutes, the performance of CGsNet decreased rapidly and then gradually decreased slowly. Although the performance is worse than that with the other two thresholds, CGsNet still exhibits forecasting skill within 2 h. Overall, the above results confirm the effectiveness of the developed CGsNet

215   model, which is skillful for nowcasting ASWS in CGs events.

**4.2   Case study of ASWS nowcasting**

To further prove the usefulness of CGsNet in detail, we show examples in two meteorologically important weather cases produced by CGsNet. The first case occurred on April 30, 2021, when eastern China was affected by cold air due to the influence of the Northeast Cold Vortex in China. This cold air accompanied by the cold vortex moved eastward and southward,

220   intersecting with some warm and humid air currents in eastern China. Then, the areas along the Yangtze River and north of Jiangsu suffered from SCW, including sudden CGs and hail. Because of the small scale and rapid development of these CGs, numerical models usually cannot forecast this process accurately.

Specifically, hail and large-scale thunderstorms occurred from 18:00 to 22:00 (BJT) in parts of Nantong, Jiangsu, with a maximum wind speed of up to 47.9 m/s in some areas. Figure 5 shows the in situ observations and forecasted ASWS during

225   this period. The CGs developed strongly at 20:12, and the ASWS exceeded 17.2 m/s in some areas. Then, the ASWS gradually decreased after 20:12 for 120 minutes. It is clear that the CGsNet model has good forecasting skill. CGsNet accurately forecasts the position of the CGs and their weakening trend. In particular, the CGsNet model can preserve the strong gust regions (ASWS>10.8 m/s) (Figure 5). The reason for this is that CGsNet employs attention schemes, which better model the short-term and long-term dependency of wind gusts. Additionally, the observed strong gusts continuously moved to the southeast

230   as the lead time continued. CGsNet forecasted this phenomenon precisely, although in the later stage, the locations of the forecasted strong gusts were not exactly the same as in the observations.

The second case is also typical convection weather. It was influenced by the low-level shear lines and strengthened by southwest warm and humid airflow on 15 May 2021. This process occurred in central and eastern Jiangsu, accompanied by heavy precipitation and strong gusts, causing serious economic losses. The observations and forecasts of ASWS for this case

235   are presented in Figure 6.

In the second case, CGs are produced with the development of the squall line in the Jiangsu area. The strong gust position is linearly distributed, moving eastward from western Jiangsu to central Jiangsu between 13:00 and 15:00 (BJT). Figure 6 clearly shows that CGsNet has good forecasting skill for the moving trend due to the improved modeling capability. Additionally, the results demonstrate that CGsNet can accurately forecast the ASWS values in most areas, although the strong gust values are

240   slightly smaller than the observations in some areas. Similar results can also be found in the RMOS forecasts (Figure A1 and A2), which are the other output variables of CGsNet. This further confirms that CGsNet is effective and accurate for ASWS nowcasting. Based on the reliable ASWS forecasts, nowcasting on the PWGS of CGs may be conducted.

[Figure]

[Figure]

[Figure]

**Figure 5.** Observations (first row) and forecasts (second row) of ASWS in eastern China, 30 April 2021, 20:12-22:12 BJT. Note that forecasting started at 20:12, and the observations and forecasts are shown at intervals of 30 min.

Anonymous: The neural network does not capture small areas of CG at all (anywhere outside of the big blob south of where I have placed this dot). There are a lot of small areas with observed wind speed in the red (> 10.8 m/s) but forecast wind speed much, much lower.

Anonymous: Also, in the large area of CG (south of where I have placed this dot), although the neural network accurately predicts strong wind (> 10.8 m/s) *kind of* in the same neighbourhood, there is still a large spatial offset between the forecast and observed strong winds. The forecast strong winds (red) are about 50 km north of the observed strong winds (red).

Anonymous: Neither of the shortcomings I pointed out in this figure are addressed in your discussion of the figure.

[revised manuscript text omitted]

---

## Author Comment (AC1)

**Response to reviewers' comments:**

**# Response to Reviewer 1**

Reviewer #1:Convective wind gusts (GCs) cause great structural damage and serious hazards. This paper designs a physics-constrained model (namely by an improved PhyDNet) for 0-2 h of quantitative CGs nowcasting with a spatial resolution of 0.01°×0.01° and a 6-minute temporal resolution. The structure of the forecast neural network is designed interesting, which contains a temporal attention module. In addition, this model combines sufficient situ observations and radar data. I admire the author's dedication to such detailed work. I only have some minor points to make before being published.

We appreciated the constructive and detailed comments from the reviewer, which helped us greatly to improve the manuscript. Please find the detailed responses to each comment below.

**[Comments]**

I  DATA

①  The wind data the authors used is from automatic weather stations (AWSs), and they are few wind observation data on the seacoast (and even over the sea) from Figure 1b. I wonder how to interpolate the wind data over the sea, whether to set it to 0 directly. Please clarify the detailed information on the IDW interpolation method.

*Response:* Thanks for your suggestion. In this study, we employed the Inverse distance-weighted (IDW) algorithm for wind data interpolation. For each grid point, we used the nearest four stations within a 15 km radius to perform the interpolation. In the case of sea areas, due to the lack of weather stations, the condition of having four stations within a 15 km radius is not met, resulting in many 'nan' values over the sea.

Additionally, since accurate wind speed observations are lacking over the sea, the interpolation in coastal areas may also have significant errors. To address this issue, we used the Natural Earth land-sea dataset to create a mask for the sea areas. We set a special value (here we set to 0) for the sea areas and assigned a weight of 0 to these regions during the training process. This ensures that the sea areas do not contribute to the backpropagation optimization.

②  If the wind data over the sea is 0, please give some analyses on whether this approach affects model performance, such as causing underestimations on PWGS to some extent.

*Response:* As mentioned above, we set the wind data over the sea as 0. During the training process, we mask the wind data on the sea areas by setting the weight of the loss function to 0. This is done to minimize the loss in forecasting gusts on land areas. Although we minimized the loss by masking the wind data during the calculation, errors still exist because the convolution calculation still involves the coastal areas. Setting the wind data on the seacoast and over the sea to 0 or a special value will not affect the model training, as the values on the seacoast and over the sea do not participate in the model's backpropagation optimization. However, during the forward propagation process, which is the forecasting process, the special values on the seacoast (and over the sea) still participate in the convolutional operations. Thus the ASWS or PWGS close to the seacoast may be slightly underestimated.

If the wind data over the sea is set to 0, but the seacoast areas are still included in the loss calculation, it could significantly impact the results and result in more pronounced underestimations.

We have added explanations for this in the revised manscript (Lines 80-81 and 86-91) : "Note that there are limited wind observation data available on the seacoast/over the sea, and as a result, this study focuses on gust forecasting in the land region of eastern China." "The wind observation data on the seacoast and over the sea was set to 0. Subsequently, during the training process, we masked the wind data on the seacoast and over the sea by setting the weight of the loss function to 0. This ensures that the sea areas do not contribute to the backpropagation optimization. However, the values in the sea areas still participate in the convolutional operations during the forward propagation process, which could lead to a slight underestimation of the ASWS or PWGS values in areas close to the seacoast."

③ The same interpolation problem also emerges in radar reflectivity. There are 10 weather radar stations in eastern China, and how to interpolate them into grided data. I suggest the authors exhibit some exemplificative image pairs for ASWS/radar data and processed gridded data.

**Response:** Thanks for your suggestion. The 3 km radar reflecticity/RMOS data was obtained from the operational Doppler weather radar 3-D digital mosaic system developed by the Chinese Academy of Meteorological Sciences (Wang et al., 2009). This system integrates data from multiple Doppler weather radar stations across eastern China, providing a comprehensive and high-resolution representation of the atmosphere at a 3 km altitude. The system can provide quality controlled base data, 3-D reflectivity grid data of single site, 3-D mosaic reflectivity and some derived products base on mosaic base data, which are useful not only for operational work, but also for scientific research.

For specific data processing, it converts the radar scan data from polar coordinates to Cartesian coordinates using a two-step interpolation process. First, it applies a nearest-neighbor interpolation method in the radial direction. Second, it uses vertical linear interpolation in the azimuthal direction.

To combine the gridded reflectivity fields from multiple radar stations, the systerm stitches them together, ensuring appropriate overlap in many regions, particularly in the middle and upper troposphere where data from multiple radars are available. After that, an exponential weighting interpolation method based on the distance between individual grid cells and radar locations is used to obtain interpolated results at an altitude of 3 km. More details of radar data interpolation can be found in (Wang et al., 2009).

We have added the mentioned radar data interpolation into the revised manuscript (Lines 105-107).

Reference:
Wang, H., Liu, L., Wang, G., Zhuang, W., Zhang, Z., and Chen, X.: Development and application of the Doppler weather radar 3-D digital mosaic system, Journal of Applied Meteorological Science, 20, 214–224, 2009 (in Chinese).

④ Is the data the authors used open-source? If so, please clarify the web links of them.
**Response:** Yes, the data we used is open-source. Please find the wind observation data and radar

data files at this site: https://doi.org/10.7910/DVN/PIZU7V. We also have added this in the "Code availability" section of the revised manuscript (Page 24).

**II  MODEL**

① Please clarify the detailed structures of the convolutional encoder and decoder in Appendix, and the feature map shapes of $h^E$, $h^p$, $h^c$, and $h^m$.

*Response:* Thanks for your detailed suggestion. The detailed structures of the convolutional encoder and decoder are shown in Table S1. The input image shape is batch_size×sequence_length×channels×width×height. Specifically, batch size is set to 2, input sequence length is 10. The feature map shapes showed in the convolutional and deconvolutional layers are just channels×width×height. After each convolutional layer in the encoder, there is a Group Normalization followed by a LeakyReLU activation function. Similarly, in the decoder, a Group Normalization and a LeakyReLU activation function follow each deconvolution layer, except for the fourth deconvolution layer.

The feature map shapes of $h^E$, $h^p$, $h^c$, and $h^m$ are both $128 \times 30 \times 35$. We also have added this in supplement of the revised manscript.

Table S1 Parameter settings in encoder and decoder. "Conv" denotes convolutional layer in encoder, "Deconv" denotes deconvolutional layer in decoder.

| Encoder | Input size | Kernel size /stride | Decoder | Input size | Kernel size /stride |
|---|---|---|---|---|---|
| Conv1 | 2×480×560 | 3×3/(2,2) | Deconv1 | 128×30×35 | 3×3/(2,2) |
| Conv2 | 16×240×280 | 3×3/(2,2) | Deconv2 | 64×60×70 | 3×3/(2,2) |
| Conv3 | 32×120×140 | 3×3/(2,2) | Deconv3 | 32×120×140 | 3×3/(2,2) |
| Conv4 | 64×60×70 | 3×3/(2,2) | Deconv4 | 16×240×280 | 3×3/(2,2) |
| Encoder output size: 128×30×35 | | | Decoder output size: 2×480×560 | | |

② What is the difference between $h^D$ and $h^m$? In Equation (4), $h^D$ is the summation of $h^p$ and $h^c$. However, in Figure 3, $h^D$ is from some transformation of $h^m$, and $h^m$ is the summation of $h^p$ and $h^c$. Please unify the illustration.

*Response:* Thanks for pointing out this. After careful checking, $h^D$ is actually $h^m$. Specifically, $h^m$ is the summation of $h^p$ and $h^c$, and it is fed into the deconvolutional units to calculate the predictions. We have corrected this mistake and modified Figure 1 (shown below) in the revised manuscript.

[Figure]

**Figure 1.** Illustration of CGsNet. The encoder is to the left of the dotted red line, and the decoder is to the right. $x_i^{a,r}$ and $\hat{x}_i^{a,r}$ are the observed and forecasted ASWS/RMOS fields, respectively. $h_i^E$ indicate the input tensors calculated by the convolution units. $h_i^c$ and $h_i^p$ indicate the hidden states of ConvLSTM and PhyCell, respectively. $h_i^m$ represents the hidden state that combines the values from $h_i^c$ and $h_i^p$.

③ From the example of Figure 3, I wonder if there are 4 individual attention modules corresponding to the output length. And the effect of the attention is to find the most significant historical information from the input sequence. For example, when predicting $t$ , the attention module can assign weights for $\{h_{t-4}^m,\ h_{t-3}^m,\ h_{t-2}^m,\ h_{t-1}^m\}$ , and when prediction $t$ + 1, the attention module can assign weights for $\{h_{t-4}^m,\ h_{t-3}^m,\ h_{t-2}^m,\ h_{t-1}^m\}$ as well, or for $\{h_{t-3}^m,\ h_{t-2}^m,\ h_{t-1}^m, h_t^m\}$.

*Response:* Yes, when predicting $\hat{x}_t^{a,r}$ , the attention module can assign weights for $\{h_{t-4}^m,\ h_{t-3}^m,\ h_{t-2}^m,\ h_{t-1}^m\}$ , and when predicting $\hat{x}_{t+1}^{a,r}$, the attention module can assign weights for $\{h_{t-3}^m,\ h_{t-2}^m,\ h_{t-1}^m,\ h_t^m\}$. We have added this explanation into the revised manscript (Lines 154-156): "The effect of the attention mechanism operation is to find the most significant historical information from the input sequences, e.g., when predicting $\hat{x}_{t+1}^{a,r}$, the attention module can assign weights for $\{h_{t-K}^m,, \ldots, h_t^m\}$."

④ There is no $k$ on the right side of Equation (1), how to obtain $s_{tk}$ ($\forall k \in [1,\ K]$)?

*Response:*Thanks for pointing out this. After careful checking, we modified Equation (1) as:

$s_{tk} = W * h_{t-k}^E + b, \forall k \in [1,\ K]$. Then $\alpha_{tk}$ is calculated by input $s_{tk}$ and it can be interpreted as the relative importance of the k-th $h^m$. Please check this in the Equation (1) of the revised manuscript.

⑤ Please give the detailed calculation process of PhyDNet and ConvLSTM in Appendix.

*Response:* Thanks for your suggestion. We have added the deailed calculation process of PhyDNet (the main module: PhyCell) and ConvLSTM in supplement, as follows:

1) PhyCell

PhyCell is a physical cell of PhyDNet (Guen and Thome, 2020b), whose dynamics are governed by the PDE response function $\mathcal{M}_p(\boldsymbol{h^p}, \boldsymbol{u})$:

$$\mathcal{M}_p(\boldsymbol{h}, \boldsymbol{u}) := \Phi(\mathbf{h}) + \mathsf{C}(\mathbf{h}, \mathbf{u}) \qquad (1)$$

where $\Phi(\mathbf{h})$ represents a physical predictor modeling only the latent dynamics, and the $\mathsf{C}(\mathbf{h}, \mathbf{u})$ represents a correction term, modeling the interaction between input data and latent state. $\Phi(\mathbf{h})$ can be modeled as:

$$\Phi(\mathbf{h}(t, x)) = \sum_{i,j:i+j \leq q} c_{i,j} \frac{\partial^{i+j} \boldsymbol{h}}{\partial x^i \partial y^j}(t, x) \qquad (2)$$

$\Phi(\mathbf{h}(t, x))$ combines the spatial derivatives with coefficients $c_{i,j}$ up to a certain differential order $q$. A wide range of classical physical models, e.g., the wave equation and heat equation, can be subsumed in this generic class of linear PDEs.

The discrete time PhyCell with the standard forward Euler numerical scheme can be written as:

$$\boldsymbol{h}(t+1) = (1 - \mathbf{K}_t) \odot \left(\boldsymbol{h}_t + \Phi(\boldsymbol{h}_t)\right) + \mathbf{K}_t \odot \mathbf{E}(\boldsymbol{u}_t) \qquad (3)$$

where $\odot$ denotes the Hadamard product; $\mathbf{K}_t$ is a gating factor; and $\mathbf{E}(\boldsymbol{u}_t)$ indicates the new observed input. Here we write the equivalent two-steps for Eq (3):

$$\widetilde{\boldsymbol{h}}_{t+1} = \boldsymbol{h}_t + \Phi(\boldsymbol{h}_t) \qquad \text{Prediction} \qquad (4)$$
$$\boldsymbol{h}_{t+1} = \widetilde{\boldsymbol{h}}_{t+1} + \mathbf{K}_t \odot (\mathbf{E}(\boldsymbol{u}_t) - \widetilde{\boldsymbol{h}}_{t+1}) \ \text{Correction} \qquad (5)$$

The Eq (4) represents the prediction step, which is a physically-constrained motion in latent space, and it computes the intermediate representation: $\widetilde{\boldsymbol{h}}_{t+1}$. The correction step in Eq (5) incorporates the input data. The decoupling between prediction and correction can be used to robustly train the model in missing data contests and long-term forecasting. Besides, the trade-off between both steps is controlled by $\mathbf{K}_t$, which can be interpreted as the Kalman gain.

The physical predictor $\Phi$ in Eq (4) is implemented by using a convolutional neural network, based on the connection between convolutions and differentiations. The 1×1 convolutions are used to linearly combine the derivatives with $c_{i,j}$ coefficients in Eq (2). Moreover, the Kalman gain $\mathbf{K}_t$ is approximated by a gate with learned convolution kernels $\mathbf{W}_h$, $\mathbf{W}_u$ and bias $\mathbf{b}_k$:

$$\mathbf{K}_t = \tanh\left(\mathbf{W}_h * \widetilde{\boldsymbol{h}}_{t+1} + \mathbf{W}_u * \mathbf{E}(\boldsymbol{u}_t) + \mathbf{b}_k\right) \qquad (6)$$

where * respent the convolutional operator. If $\mathbf{K}_t = 0$, the dynamic follows the physical predictor; if $\mathbf{K}_t = 1$, the latent will be reset and only driven by the input.

PhyCell is an atomic recurrent cell used for building physically-constrained RNNs. The PhyDNet here uses one layer of PhyCell, which can also be easily stacked to build more complex models.

2) ConvLSTM

ConvLSTM is a variant of the long short-term memory network (Shi et al., 2015), which is a fundamental and effective spatiotemporal recurrent structure for spatiotemporal modeling. The ConvLSTM uses the forget gate, input gate, and output gate to update its cell and hidden states. The input gate controls how much of the new information will be added to the memory cell. The

forget gate is used to control how much of the previous information will be forgotten from the memory cell, while the cell information which will be propagated to the new gate is controlled by the output gate. The calculation processes of the ConvLSTM are as follows:

$$i_t = \sigma(W_{xi} * x_t + W_{hi} * h_{t-1} + W_{ci} \odot c_{t-1} + b_i) \tag{7}$$

$$f_t = \sigma(W_{xf} * x_t + W_{hf} * h_{t-1} + W_{cf} \odot c_{t-1} + b_f) \tag{8}$$

$$c_t = f_t \odot c_{t-1} + i_t \odot tanh(W_{xc} * x_t + W_{hc} * h_{t-1} + b_c) \tag{9}$$

$$o_t = \sigma(W_{xo} * x_t + W_{ho} * h_{t-1} + W_{co} \odot c_t + b_o) \tag{10}$$

$$h_t = o_t \odot \tanh(c_t) \tag{11}$$

where $\sigma$ is the sigmoid activation function. Besides, $x_t$, $c_t$, $h_t$, $i_t$, $f_t$, and $o_t$ represent input, memory cell, hidden state, input gate, forget gate, and output gate, respectively. Bias $b$ and weight $W$ both represent learning parameters.

References:

Guen, V. L. and Thome, N.: Disentangling physical dynamics from unknown factors for unsupervised video prediction, in: Proceedings of the IEEE/CVF Conference on Computer Vision and Pattern Recognition, pp. 11 474–11 484, https://doi.org/10.1109/CVPR42600.2020.01149, 2020b.

Shi, X., Chen, Z., Wang, H., Yeung, D.-Y., Wong, W.-K., and Woo, W.-c.: Convolutional LSTM network: A machine learning approach for precipitation nowcasting, Advances in neural information processing systems, 28, https://doi.org/10.1007/978-3-319-21233-3_6, 2015.

⑥ About the physics-constraint concept mentioned in Introduction and Model architecture (namely PhyDNet), I think the authors should cite the "inductive biases" from doi.org/10.1038/s42254-021-00314-5, which can be interpreted by designing specialized network architectures that implicitly embed prior knowledge and satisfy a set of given physical laws.

*Response:* Thanks for your suggestion. We have cited the "inductive biases" in the Model architecture section: "PhyDNet steers the learning process toward identifying physically consistent solutions by introducing an appropriate inductive bias (Karniadakis et al., 2021), that is, implicitly embedding prior knowledge in the network architecture and satisfying a given set of physical laws." (see Lines 126-128 in revised manuscript).

Reference:

Karniadakis, G.E., Kevrekidis, I.G., Lu, L. et al.: Physics-informed machine learning, Nature Reviews Physics, 3, 422–440, https://doi.org/10.1038/s42254-021-00314-5, 2021.

III  **EXPERIMENT**

① Please append an ablation study on the proposed attention module.

*Response:* Thanks for the suggestion. As suggested, we have added an ablation study on the attention module. Specifically, we compared the performance of PhyDNet (without attention) with CGsNet. Table 2, Figure 4-6 show the results of the ablation study, which indicate that the

forecasting performance of CGsNet for ASWS is superior to that of PhyDNet. These results suggest that the attention mechanism proposed in CGsNet is effective and can significantly improve the accuracy of ASWS forecasting. Overall, the results confirm that CGsNet is reliable and accurate for ASWS nowcasting. Detailed comparative descriptions can be found in Sections 4.1 and 4.2 of the revised manuscript.

**Table 2.** Quantitative results of CGsNet and PhyDNet on ASWS nowcasting. 95% CI represent the 95% confidence interval of the indices.

| Model | ASWS (m/s) | CSI 95% CI | HSS 95% CI | POD 95% CI | MAE 95% CI | RMSE 95% CI |
|---|---|---|---|---|---|---|
| **CGsNet** | **8.0** | 0.41 (0.33; 0.49) | 0.54 (0.47; 0.61) | 0.59 (0.51; 0.66) | 1.60 (1.46; 1.80) | 2.26 (2.00; 2.54) |
| | **10.8** | 0.31 (0.22; 0.40) | 0.42 (0.32; 0.50) | 0.47 (0.34; 0.60) | 2.19 (1.93; 2.51) | 3.07 (2.62; 3.56) |
| | **13.9** | 0.15 (0.04; 0.21) | 0.20 (0.07; 0.24) | 0.22 (0.10; 0.31) | 2.90 (2.26; 2.66) | 4.22 (3.25; 5.10) |
| **PhyDNet** | **8.0** | 0.39 (0.31; 0.47) | 0.52 (0.44; 0.59) | 0.55 (0.46; 0.62) | 1.71 (1.55; 1.91) | 2.40 (2.14; 2.68) |
| | **10.8** | 0.28 (0.20; 0.38) | 0.38 (0.28; 0.47) | 0.41 (0.29; 0.53) | 2.40 (2.12; 2.76) | 3.33 (2.90; 3.83) |
| | **13.9** | 0.12 (0.03; 0.19) | 0.16 (0.05; 0.21) | 0.19 (0.09; 0.28) | 3.10 (2.39; 3.97) | 4.54 (3.59; 5.47) |

[Figure]

**Figure 4**. The CGsNet and PhyDNet results for different nowcasting lead times of ASWS at thresholds of 8.0 m/s, 10.8 m/s, and 13.9 m/s.The shaded red and green areas represent the 95% confidence intervals of the CGsNet and PhyDNet indices, respectively.

[Figure]

**Figure 5.** Observations (first row), CGsNet forecasts (second row), and PhyDNet forecasts (third row) of ASWS in eastern China, 30 April 2021, 20:12-22:12 BJT. Note that forecasting started at 20:12, and the observations and forecasts are shown at intervals of 30 min.

[Figure]

**Figure 6.** Same as Figure 5, but for 15 May 2021, 13:00-15:00 BJT.

**Response to reviewers' comments:**

**# Response to Reviewer 2**

Reviewer #2:This paper describes the use of a neural network for predicting convective wind gusts at lead times of 6-120 minutes. The neural network includes several architectural components that have recently been developed in the field of deep learning. These include pieces of the recurrent neural net (RNN) architecture, where the prediction at the $k^{th}$ lead time becomes an input (predictor) to the neural network to predict at the $(k + 1)^{th}$ lead time; the attention mechanism, where the neural network can automatically determine the most important steps in a time series of predictor values; and PhyCell, which incorporates partial differential equations (PDE). The neural network predicts convective wind gusts on a 1-km grid covering eastern China. The authors compare their results to a baseline model called INCA and demonstrate (albeit without significance-testing) that their neural network, which they call CGsNet, outperforms INCA. Overall, the quality of both the writing and science are very good. I have a handful of major comments, which are made as inline comments in the attached PDF but also summarized below. Minor comments are not summarized below and can be seen only in the attached PDF.

Thanks for the constructive and detailed comments, which help us improve the manuscript significantly. We considered all comments and carefully revised all comments in the revised manuscript. Below is the point-to-point response to major and minor comments.

**[Major comments]**

1.   The abstract contains several unclear/unjustified statements. See inline comments.

***Response:***Thanks for your detailed suggestion. We have revised the abstract section, addressing previously unclear or unjustified statements. Please check abstract section in the revised manuscript. About several comments, the responses are as follow:

1)   Line 5: Do you mean "at 0--2-hour lead time"?

Yes. We have modified this and made it clear (see Line 5).

2) Line 5: What do you mean by "first"?   That your approach is the first ever to achieve minute-/km-level forecasts?

In fact, accurately forecasting CGs at the minute-kilometer-level has been a challenge, with few past studies achieving this goal. While fine-scale numerical experiments can predict CGs at this level, they often require long computation times on the order of several hours, making them unsuitable for nowcasting needs. Therefore, we have removed the word "first" in the revised manuscript for the sake of accuracy.

3) Line 7: What do you mean by "spatiotemporally consistent"?   This was never discussed in the paper.

The "spatiotemporally consistent" means that the CGs forecasts are continuous in temporal and spatial within the 0-2 hour forecast window. Unlike traditional extrapolation-based forecasting algorithms such as SCIT (Storm Cell Identification and Tracking) (Johnson et al., 1998), which focus on convective cells and require identification of these cells before issuing extrapolation forecasts and rough early warnings for affected areas, our algorithm can perform 2D grid-based forecasts with 6-minute updates in all weather conditions. Therefore, we describe our algorithm as

spatiotemporally consistent.

(a) Calculating the distance $d$ from unknown points to all points;
(b) Calculating the weight $\lambda$ for each point, the weight is a function of the reciprocal of the distance: $\lambda_i = \dfrac{d_i^{-p}}{\sum\limits_{i=1}^{K} d_i^{-p}}$, where $K$ is the total number of discrete points. Power ($p$) is a parameter used to calculate the influence weight of the nearest $K$ discrete points on the interpolation site, which controls the effect of known points on the interpolation value based on their distance from the interpolated point.

(c) Calculating interpolation results: $\hat{z}(x_0, y_0) = \sum\limits_{i=1}^{K} \lambda_i z(x_i, y_i)$, where $(x_0, y_0)$ is the interpolation point coordinate and $(x_i, y_i)$ represents the coordinates of discrete points.

To determine the optimal parameters for the IDW interpolation, we experimented with various combinations of the number of stations (K) and radius of influence (R) (Figure S13). In IDW interpolation, R refers to finding the K nearest discrete points from the interpolation site within a radius range of R. As shown in the Figure S13, it is evident that when R is set to 5 km, the condition of having nearest discrete points/stations within a 5 km radius is not met, resulting in many 'nan' values. This is because the distance between meteorological observation stations is 10 km. Therefore, when R is less than 10 km, the observed field cannot be adequately interpolated.

However, when R is set to 10 km or lager, the interpolation results do not significantly change. For K, when taking different values ranging from 2 to 16, the interpolation results are comparable. As K increases, the value of the interpolated point is progressively smoothed by the surrounding discrete points. Generally, when the value of R is not less than 10 km and the value of K is not too large, the interpolation results of the observed wind field are similar, and the results are effective. Thus using a R of 15 km and K= 4 staions are feasible. The choice of 15 km radius of influence allowed us to capture local variations in the data without being overly influenced by distant stations, which might not have the same meteorological conditions. Using 4 stations ensured that the interpolation incorporated sufficient data points.

For power, as the power value increases, the interpolated value will gradually approach the value of the nearest point. By specifying a small power value, the influence of points farther away will be great, resulting in a smooth plane. Based on a comprehensive analysis, we selected a common value of power=2. The power of 2 provided a balance between the weighting of nearby and distant stations, which has been widely used in many studies.

[Figure]

**Figure S13.** The various combinations of the number of stations (K) and radius of influence (R) for IDW interpolation.

**(2)For the lines 150-152:**

To ensure the robustness and generalization ability of our model, we carefully selected the hyperparameters through a priori reasoning. The reasons for these hyperparameter choices are as follows:

①Batch size: the batch size of 2 was chosen based on two factors. First, a small batch size also has good generalization ability since each batch is randomly selected, allowing the model to adapt better to new data. Second, a small batch size helps to reduce memory usage and accelerate training, as the samples occupy a significant amount of memory.

②Learning rate: the learning rate was set based on the settings of PhyDNet (Guen and Thome, 2020b), which involved selecting an initial learning rate of 0.001 and decreasing it if the loss function did not decrease after several epochs of training.

③Epoch: setting the number of epochs to 50 is to ensure that the model is fully trained. When saving the model, the one with the lowest validation loss was selected and saved.

**(3)Regarding Table 1:**

The thresholds parameters of loss function was mainly based on the wind speed classification determined by the China Meteorological Administration (https://www.cma.gov.cn/2011xzt/20120816/2012081601/201208160101/201407/t20140717_252 607.html ), as shown in the following Table S1. High weights were assigned to high wind speeds and RMOS values since accurate predictions in these ranges are crucial for ensuring life safety and minimizing potential damage caused by extreme weather events. For example, wind speed greater than 20.8 m/s are considered to be particularly hazardous, and thus, predictions in this range were given a high weight. Besides, the issue of imbalance between high wind speed data and low wind speed data can be partially addressed by assigning different weights, as there is a scarcity of high wind speed data samples and an abundance of low wind speed data samples. This approach can help alleviate the problem of the insufficient number of high wind speed data samples and enable the model to forecast areas where strong gusts occur, which is a significant concern. The specific weights were set with reference to the settings of Shi et al (2015) and expert advice.

Table S1. The wind speed levels defined in China

| Wind level | Wind speed (m/s) |
| --- | --- |
| 0 | [0.0-0.3) |
| 1 | [0.3-1.6) |
| 2 | [1.6-3.4) |
| 3 | [3.4-5.5) |
| 4 | [5.5-8.0) |
| 5 | [8.0-10.8) |
| 6 | [10.8-13.9) |
| 7 | [13.9-17.2) |
| 8 | [17.2-20.8) |
| 9 | [20.8-24.5) |
| …… | |

*Response:*Thanks for the suggestion. We have added this into the revised manuscript in line 441.